# Toward Linearly Regularizing the Geometric Bottleneck of Linear Generalized Attention

**Jiaxu Liu**[1], **Xinping Yi**[2*], **Xiangyu Yin**[1], **Yuhang Song**[1,3], **Gaojie Jin**[4], **Xiaowei Huang**[1]
*{jiaxu.liu, x.yin22, sgyson10, xiaowei.huang}@liverpool.ac.uk, xyi@seu.edu.cn, gaojie.jin.kim@gmail.com*
[1] *University of Liverpool,* [2]*Southeast University,* [3]*National Tsing Hua University,* [4]*University of Exeter*

**Reviewed on OpenReview:** *https://openreview.net/forum?id=Vpyg3fqXbl*

## Abstract

Transformers excel across domains, yet their full self-attention carries a prohibitive $\mathcal{O}(n^2)$ cost for long sequences with length $n$. Existing *efficient* attention methods either restrict the attention pattern (local/sparse attention) or approximate the softmax kernel with certain drawbacks. The former suffers from attention bottlenecks (over-squashing of long-range dependencies) and invalidates the use of global tokens in autoregressive tasks, while the latter often requires sequential processing that can degrade in accuracy when approximations fall short. In this work, we introduce the *Bottleneck Regularized Linear Attention (BRL-Attention)*, uniting the strengths of pattern-based and kernel-based techniques to enable efficient, global information flow with linear complexity. BRL-Attention extends a local attention pattern with a small set of compressed tokens that serve as a global information reservoir, ensuring long-range interactions without quadratic cost. This bottleneck regularization strategy effectively alleviates the geometric attention bottleneck and retains full expressiveness; that is, it matches the sequence modeling capacity of full softmax attention while mitigating over-squashing across layers. Moreover, it integrates global tokens without breaking causal masking, making it applicable to both encoder-only and autoregressive decoder architectures. Extensive experiments on sequence and graph benchmarks demonstrate that BRL-Attention matches or surpasses the predictive performance of standard Transformers with full attention, while substantially reducing memory usage and computation time to levels comparable with linear sparse attention.

Transformers (Vaswani et al., 2017) have substantially advanced the state-of-the-art in areas such as natural language processing, computer vision, and graph learning (Dosovitskiy et al., 2020; Touvron et al., 2021; Devlin et al., 2018; Touvron et al., 2023). Their core strength lies in the attention mechanism, which models global token-to-token interactions. However, *full* self-attention involves computing pairwise relationships among all tokens, leading to a formidable $\mathcal{O}(n^2)$ time and memory complexity, given the sequence length of $n$. This limitation becomes a major obstacle for tasks involving long sequences or large-scale datasets, where computational overhead can escalate dramatically.

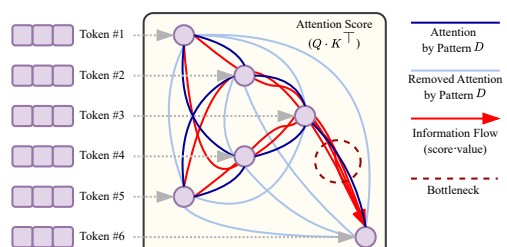

Figure 1: Visualization of *Attention Bottleneck* induced by pattern-based generalized attention. Information of token #1 and token #5 struggle to propagate to token #6 within one self-attention block.

In response, various *efficient transformers (ET)* have been proposed to replace full attention with mechanisms that cost at most linear time in the sequence length (Tay et al., 2022). These can be broadly categorized into *pattern-based* and *kernel-based* methods (Sec. 1.1). Pattern-based transformers (Sec. 1.1),

*Corresponding author

such as Sparse Transformers (Child et al., 2019), Longformer (Beltagy et al., 2020), ETC (Ainslie et al., 2020), and BigBird (Zaheer et al., 2020), restrict each query token to attend to a local or blockwise subset of the entire sequence, reducing complexity to near $\mathcal{O}(n)$ by exploiting structured sparsity. Despite their efficiency, these localized patterns can cause an *attention bottleneck* (*a.k.a. over-squashing* (Alon & Yahav, 2020; Topping et al., 2021)), as illustrated in Fig. 1, where distant tokens fail to interact effectively within the limited receptive fields of each layer. While introducing global tokens or memories can alleviate this for encoder-only tasks, it often breaks causal masking and thus remains impractical for autoregressive decoding. In contrast, *kernel-based* transformers (Sec. 1.1) approximate the softmax via low-rank projections or random feature maps (Katharopoulos et al., 2020; Choromanski et al., 2020), also aiming for linear or near-linear time. However, these methods can degrade performance if the chosen approximation rank is insufficient and often struggle in autoregressive settings: the need for causal masking typically forces sequential attention computation, forfeiting any parallelizable speedups. They also tend to be sensitive to random-feature variance, especially in scenarios involving domain shifts or noise.

In this work, we propose a new paradigm, *Bottleneck Regularized Linear Attention (BRL-Attention)*, that combines the advantages of pattern- and kernel-based approaches while sidestepping their key limitations (discussed in Rmk. B.1). BRL-Attention extends any sparse/pattern-based attention mechanism with a small set of *compressed tokens*, which serve as a global information reservoir. These tokens can be integrated without invalidating causal structure in autoregressive tasks. We formalize this design as a *bottleneck regularizer* (Sec. 2.3) that channels distant dependencies through these compressed tokens. Particularly, we introduce two key functions to facilitate an expressive attention, *Compression*

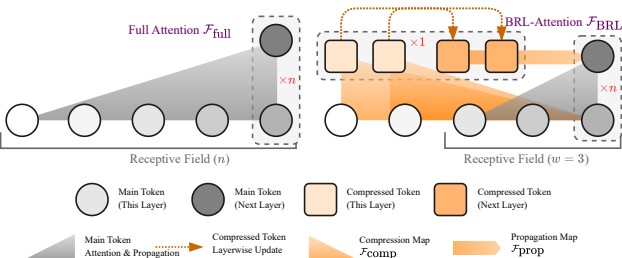

Figure 2: (**left**) Quadratic Full-Attention; (**right**) Proposed BRL-Attention (Linear) with a window/chunk size ($w$) of 3 ($w$ is typically set to $\geq 64$ in our experiments).

and *Propagation*, which are two communication functions defined between main tokens and compressed tokens. By design, these two functions help alleviate the attention bottleneck induced by over-squashed patterns, while offering benefits *e.g.*, preserving the expressibility and robustness on noisy attention. While *linearizing* attentions inevitably sacrifices expressibility (Hua et al., 2022), we theoretically prove the comparable expressibility of BRL-Attention towards full-attention, while mitigating over-squashing between contiguous layers. The resulting approach scales linearly with the sequence length on inference time, memory, and autoregressive training time.

Our main contributions are highlighted as follows: *(1)* Starting with a *sensitivity analysis* (Sec. 2.2), we demonstrate that conventional sparse patterns can fail to propagate long-range information within a small number of layers. In contrast, adding our bottleneck regularizer effectively recovers global context with minimal overhead. *(2)* We propose the *Bottleneck-Regularized Linear Attention (BRL-Attention)* mechanism, which augments any sparse/pattern-based attention with compressed tokens. This mechanism ensures efficient $\mathcal{O}(n)$ complexity and alleviates information bottlenecks in strictly local attention. Theoretically, we justify that with all introduced techniques in BRL-Attention, the resulting *BRL-Former* facilitates a wider sensitivity bound between distant tokens in intersective layers (Sec. 2.4), which helps alleviate attention bottleneck. Meanwhile, we justify in Sec. 2.3-2.4 that BRL-Former is as expressive as the full-attention-transformer. *(3)* Through extensive experiments (Sec. 3), including long-sequence modeling and large-graph node classification, we show that BRL-Attention not only matches or surpasses full-attention transformers but also substantially reduces memory usage and computational cost.

# 1 Preliminary

**Definition 1.1** (Generalized Attention Mechanism (Zaheer et al., 2020))**.** Given the input token to the layers $\mathbf{x} \in \mathbb{R}^{n \times d}$ with sequence length $n$ and embedding dimension $d$, the generalized attention mechanism is described by a directed graph $D$ (*a.k.a* Attention Pattern (Tay et al., 2022)) whose vertex set is $\mathcal{V} = \{1, ..., n\}$. The set of arcs (i.e., directed edges) represent the set of inner products that the attention mechanism will

Figure 3: (**left**) Sliding Window (SW) pattern $D_{\text{sw}}$; (**mid**) Visualization of graph under different configurations of window sizes $w$; (**right**) Various attention pattern $D$ employed in literature, where grey areas are either scores for individual token-pairs or sub-block of full attention.

consider. Let $\mathcal{N}_D(i)$ denote the out-neighbors set of node $i$ in $D$, then the $i$-th output vector at layer $l \in [1, L]$ of the generalized attention mechanism is defined as Eq. (1)

$$\mathcal{F}_{\text{gen}}^{(l)}(\mathbf{x}^{(l)}; D)_i = \sum_{h=1}^{H} \sum_{j \in \mathcal{N}_D(i)} \frac{\kappa(\mathbf{q}_i^{(l)}, \mathbf{k}_j^{(l)})}{\sum_{k \in \mathcal{N}_D(i)} \kappa(\mathbf{q}_i^{(l)}, \mathbf{k}_k^{(l)})} \mathbf{v}_j^{(l)} \quad (1) \qquad \mathcal{F}_{\text{full}}^{(l)}(\mathbf{x}^{(l)})_i = \sum_{j=1}^{n} \frac{\kappa(\mathbf{q}_i^{(l)}, \mathbf{k}_j^{(l)})}{\sum_{k=1}^{n} \kappa(\mathbf{q}_i^{(l)}, \mathbf{k}_k^{(l)})} \mathbf{v}_j^{(l)} \quad (2)$$

where $\mathbf{q} = \mathbf{x}\mathbf{W}_q$, $\mathbf{k} = \mathbf{x}\mathbf{W}_k$ and $\mathbf{v} = \mathbf{x}\mathbf{W}_v$ are query, key, values and $\mathbf{W}_* \in \mathbb{R}^{d \times d}$ are learnable projection weights. We use $\kappa : \mathbb{R}^d \times \mathbb{R}^d \to \mathbb{R}$ to denote the softmax kernel $\kappa(\mathbf{a}, \mathbf{b}) := \exp(\mathbf{a}\mathbf{b}^\top)$ (where $\mathbf{a}, \mathbf{b} \in \mathbb{R}^{1 \times d}$).

For brevity, we state the results with the batch size $B = 1$ and the number of heads $H = 1$ without loss of generality. If $D$ is the complete graph with adjacency matrix $\mathbf{S}$, we recover the full attention mechanism of $\mathcal{O}(n^2)$ complexity (Vaswani et al., 2017), expressed as in Eq. (2).

## 1.1 Efficient Transformers

Existing works seek to improve memory efficiency in transformers through weight pruning (Michel et al., 2019), weight factorization (Lan et al., 2019), weight quantization (Zafrir et al., 2019), efficient pretraining (Clark, 2020), attention optimization (Lample et al., 2019), or knowledge distillation. Reducing the memory or computational requirements with these methods leads to training or inference time speedups, but fundamentally, the time complexity is still quadratic *w.r.t.* the sequence length which hinders scaling to long sequences. In this paper, we mainly focus on the two lines of works that achieves near **linear complexity** transformers, namely, the pattern-based approach and kernel-based approach. We defer a more detailed discussion on related works to Appendix D.4.

**Pattern-Based Partial Attention.** Pattern-based efficient transformers mitigate the quadratic complexity of evaluating $\mathcal{F}_{\text{full}}$ by imposing structured sparsity patterns. Early methods, such as Sparse Transformers (Child et al., 2019), employ blockwise or strided attention mechanisms to reduce computations, achieving complexity of $\mathcal{O}(n\sqrt{n})$. Longformer (Beltagy et al., 2020) and ETC (Ainslie et al., 2020) further optimize this by combining local attention windows with global memory tokens at custom locations, resulting in complexity of $\mathcal{O}(wn)$, where $w$ is the window size. More of this category includes axial (Ho et al., 2019), learnable patterns through hashing (Kitaev et al., 2020) or clustering (Roy et al., 2021). These approaches strike a balance between efficiency and coverage, making them ideal for long-sequence tasks. However, one key problem with this class of methods is that they involve blocking operations, which are not parallel-friendly and could induce potential *attention bottlenecks*, as discussed in Sec. 2.2.

**Kernel-Based Linear Attention.** Kernel-based attentions approximate self-attention to achieve linear complexity. Specifically, according to Mercer's theorem (Mercer, 1909; Aizerman, 1964), the eigenfunctions corresponding to the non-zero eigenvalues are continuous on $\mathbb{R}^d$ and $\kappa$ can be represented as $\kappa(\mathbf{x}, \mathbf{y}) = \sum_{i=1}^{\infty} \lambda_i \phi(\mathbf{x}_i) \phi(\mathbf{y}_i)$. With such a property, the kernelized-full attention is typically given as

$$\mathcal{F}_{\text{kernel}}^{(l)}(\mathbf{x}^{(l)})_i = \frac{\phi(\mathbf{q}_i^{(l)}) \sum_{j=1}^{n} \phi(\mathbf{k}_j^{(l)})^\top \mathbf{v}_j^{(l)}}{\phi(\mathbf{q}_i^{(l)}) \sum_{k=1}^{n} \phi(\mathbf{k}_k^{(l)})^\top}, \text{ where } \kappa(\mathbf{x}, \mathbf{y}) = \langle \Phi(\mathbf{x}), \Phi(\mathbf{y}) \rangle_{\mathcal{V}} \approx \phi(\mathbf{x})\phi(\mathbf{y})^\top. \quad (3)$$

With certain error gap, Eq. (3) essentially tells $\mathcal{F}_{\text{kernel}}^{(l)}(\mathbf{x}^{(l)}) \approx \mathcal{F}_{\text{full}}^{(l)}(\mathbf{x}^{(l)})$. With such an idea, approaches *e.g.* (Choromanski et al., 2020; Katharopoulos et al., 2020; Wang et al., 2020; Peng et al., 2021) reduce the quadratic cost of $\mathcal{F}_{\text{full}}$ to roughly $\mathcal{O}(\max(n, d^2))$. Such kernel-based approximations eliminate the need to compute or store full attention matrices, making them well-suited for resource-constrained scenarios and real-time processing. However, kernel-based methods may exhibit limitations in parallelizing across the time

dimension during training in an autoregressive teacher forced setting. As a result, there exists a considerable gap between the theoretical complexity and actual running time. Similar to the findings in (Hua et al., 2022), we find that directly computing the full quadratic attention matrix is even faster than the kernal-based approaches on GPUs (see Sec. 3.1).

## 2 Method

In this paper, we present a new attention paradigm replacing conventional full attention, namely, the Bottle-neck Regularized Linear Attention (BRL-Attention). We employ the notation of $\mathcal{F}$ as matrix-valued function and $f$ as vector-valued function. The transformer block with BRL-Attention is formulated as

$$\mathcal{F}_{\text{BRL}}^{(l)}(\mathbf{x}^{(l)}) = \underbrace{\mathcal{F}_{\text{gen}}^{(l)}(\mathbf{x}^{(l)}; D)}_{\text{Generalized Attn}} + \underbrace{\mathcal{F}_{\text{prop}}^{(l)}(\mathbf{x}^{(l)}; \mathbf{x}_{[\text{ct}]}, \lambda)}_{\mathcal{O}(n) \text{ Regularizer}}, \tag{4}$$

$$\mathbf{x}_{\text{out}}^{(l)} = \text{LN}(\mathcal{F}_{\text{BRL}}^{(l)}(\mathbf{x}^{(l)})) + \mathbf{x}^{(l)}, \quad \mathbf{x}^{(l+1)} = \text{LN}(f_{\text{w2}}^{(l)}(\sigma_{\text{relu}}(f_{\text{w1}}^{(l)}(\mathbf{x}_{\text{out}}^{(l)})) \odot f_{\text{w3}}^{(l)}(\mathbf{x}_{\text{out}}^{(l)}))) + \mathbf{x}_{\text{out}}^{(l)}. \tag{5}$$

The key difference of $\mathcal{F}_{\text{BRL}}$ to $\mathcal{F}_{\text{gen}}(\cdot; D)$ is the introduction of regularization term $\mathcal{F}_{\text{prop}}^{(l)}(\cdot)$, which is a *propagation* of compressed information $\mathbf{x}_{[\text{ct}]}$, regarded as a patch to $\mathcal{F}_{\text{gen}}^{(l)}(\cdot; D)$ that alleviate over-squashing. The equations in Eq. (5) are standard transformer layers with skip-connection and SwiGLU (Shazeer, 2020) where $\odot$ is the Hadamard product, $\sigma_{\text{relu}}(\cdot)$ is the ReLU activation and LN denotes LayerNorm.

As below, we outline the core components and theoretical foundations of our approach. *(1)* We first establish the necessity of imposing a structured attention pattern $D$ to achieve linear complexity, and demonstrate how certain sparse patterns in $\mathcal{F}_{\text{gen}}(\cdot; D)$ can induce *attention information bottlenecks*, wherein token-level interactions fail to propagate effectively across limited transformer depths. *(2)* We then introduce the bottleneck regularizer $\mathcal{F}_{\text{prop}}$, constructed via a set of compressed tokens $\mathbf{x}_{[\text{ct}]}$. Despite its linear complexity, we show that $\mathcal{F}_{\text{prop}}(\cdot; \mathbf{x}_{[\text{ct}]}, \lambda)$ maintains an expressiveness comparable to that of kernelized attention mechanisms. *(3)* Next, we theoretically justify that, when the compression mapping $\mathcal{F}_{\text{comp}}$ is appropriately instantiated, switching the Full-Attention $\mathcal{F}_{\text{full}}$ to the BRL-Attention $\mathcal{F}_{\text{BRL}}$ results in a broader sensitivity bound. With our customization, $\mathcal{F}_{\text{BRL}}$ alleviates over-squashing and improves robustness to noisy attention weights. *(4)* Finally, under any arbitrary sparse attention pattern $D_*$, we prove that with a trivial per-head regularization coefficient $\lambda = \{\mathbf{1}\}_H$, the BRL-Attention operator $\mathcal{F}_{\text{BRL}}$ remains provably as expressive as $\mathcal{F}_{\text{full}}$ (see Thm. 2.9). Collectively, these results establish BRL-Attention as a linear-complexity attention mechanism that retains the full expressive capacity of standard transformers while offering improved scalability for long-sequence tasks. *(* All proofs are deferred to Appendix E)*.

### 2.1 Necessity of Pattern $D$ to Linear Generalized Attention

As detailed in Sec. 1.1 and (Tay et al., 2022), the pattern $D$ that facilitates $\mathcal{O}(n)$ attention essentially encompasses *blockwise* and *strided* pattern attention. The blockwise approaches (Qiu et al., 2019) chunk input sequences into blocks that reduces the complexity from $n^2$ to $n_{\text{block}}^2$ where $n_{\text{block}} \ll n$, then the $n \times n$ score is computed intermediately by computing and combining divided $n_{\text{block}} \times n_{\text{block}}$ blocks, selectively picking blocks leads to linear complexity. For the strided attention patterns (Beltagy et al., 2020; Ainslie et al., 2020) approach, we illustrate in Fig. 3 a typical pattern $D$, *i.e.* the sliding window pattern, which is a fixed-size window mask surrounding each token. Given a fixed window size $w$, each token attends to $w/2$ tokens on each side. As show in Fig. 3(left), as only gray areas are need for computation, we can essentially group the gray tokens/blocks as $n \times w$ size matrix, resulting in a $\mathcal{O}(wn)$ complexity score computation. (Zaheer et al., 2020) further combined the blockwise and strided approach, facilitating the block-diagonal, block-window local, and block-random patterns for efficient and expressive linear attention.

### 2.2 Sensitivity Analysis of Generalized Attention

As the layer-wise attention $\mathbf{x}_i^{(l+1)} \leftarrow \mathcal{F}_{\text{gen}}^{(l)}(\mathbf{x}^{(l)}; D)_i$ is continuous and differentiable, the bottleneck of attention information can then be understood in terms of one token embedding $\mathbf{x}_i^{(l+1)}$ failing to be affected by

another (previous layer-) feature $\mathbf{x}_p^{(*)}$ of token $p$ at distance $M$ from node $i$. Hence, we employ the Jacobian $\partial \mathcal{F}^{(l)}/\partial \mathbf{x}_p^{(*)}$ as an explicit and formal way of assessing the bottleneck-ed attention.

**Definition 2.1** (Attention Bottleneck)**.** Under the definition of generalized attention, $\mathcal{F}_{\text{gen}}(\cdot; D)$ with a particular $D$ within $l$ layers is said to be **bottlenecked** when there exists a token pair $i \sim p$ such that $\partial[\mathcal{F}_{\text{gen}}^{(l)}(\mathbf{x}^{(l)}; D)]_i/\partial \mathbf{x}_p^{(0)} \approx 0$. This means no information is flow from $i$ to $p$ with $l$ layers of attention blocks. If $l$ is nearly $L$, then the whole transformer suffers information bottleneck similar to MPNNs.

**[Theory]** *Failing Cases Under Certain Patterns* $D$. We reveal theoretically that the generalized attentions suffer from attention bottleneck with some patterns defined by certain $D$. Recall strided sliding window empowered linear transformers according to Sec. 2.1. Using multiple stacked layers of such windowed attention results in a receptive field, so that top layers have access to a board range of input locations and have the capacity to build representations that incorporate information across the input, similar to MPNNs. The computation complexity of this pattern is $\mathcal{O}(wn)$, which scales linearly with input sequence length $n$. Now, letting the adjacency matrix derived from pattern $D_{\text{sw}}$ be $\mathbf{S}^{\text{sw}}$, we state the following proposition.

**Proposition 2.2.** *Let $D = D_{\text{sw}}$ where the sliding window size is $w$ and $\mathbf{S}^{\text{sw}}$ has eigenvalues bounded by $r_{\text{sw}}$. Considering the source token $i$ and target token $p$ that are $M$ distance away, we have the sensitivity bound*

$$\left\| \frac{\partial \mathcal{F}_{\text{gen}}^{(l)}(\cdot; D_{\text{sw}})_i}{\partial \mathbf{x}_p^{(0)}} \right\| \leq \begin{cases} 0 & if \ l < \lceil \frac{2M}{w-1} \rceil - 1 \ , \\ (r_{\text{sw}} r_W)^{l+1} & otherwise. \end{cases} \tag{6}$$

Prop. 2.2 indicates the particular $D = D_{\text{sw}}$ leads to a *squashed* information propagation, *i.e.* for the top layer token $i$'s embedding to perceive token $p$, we need at least $\lceil \frac{2M}{w-1} \rceil - 1$ general attention layers.

*Remark* 2.3. We regard $D_{\text{sw}}$ as a simple example to illustrate how certain restricted $D$ induces substantial bottlenecks in attention propagation. In fact, many other $D$ with limited connectivity and sparse interaction patterns (*e.g.* diag chunk, dilated sliding window) exhibit similar issues, where the Jacobian with respect to distant tokens becomes effectively negligible at intermediate layers. This makes the attention mechanism behave analogously to a communication-limited message passing network, which hinders the model's capacity to integrate information across long sequences.

## 2.3 The Bottleneck Regularized Linear Attention

This section tackles the RQ: *How can a small, trainable set of global memory capture long-range context in linear complexity while preserving the expressibility.* We investigate it by *(1)* introducing the compressed tokens that evolve by layer to accumulate global summaries, and then *(2)* coupling them to the main tokens through a bottleneck regularizer $\mathcal{F}_{\text{prop}}$ that propagate the reserved information context embedding, thereby marrying linear-attention efficiency with full-attention reachability.

**[Practice]** *Compressed Token as Information Reservoir*. We first introduce a key concept that facilitates the bottleneck regularizer, namely, the compressed token. Given $m$ as the sequence length of the compressed token and $d_{\text{ct}}$ as the embedding dimension, we represent the compressed tokens as $\mathbf{x}_{\texttt{[ct]}} \in \mathbb{R}^{m \times d_{\text{ct}}}$. The layer-wise update process of the **trainable $\mathbf{x}_{\texttt{[ct]}}$** is then defined as

$$\mathbf{x}_{\texttt{[ct]}}^{(l+1)} = \sigma_{\text{relu}}((1 - \beta)\text{LN}(\mathcal{F}_{\text{comp}}^{(l)}(\mathbf{x}_{\texttt{[ct]}}^{(l)}; \mathbf{x}^{(l)})) + \beta \mathbf{x}_{\texttt{[ct]}}^{(l)} \mathbf{M}^{(l)}) \in \mathbb{R}^{m \times d_{\text{ct}}}, \tag{7}$$

where the initialization $(\mathbf{x}_{\texttt{[ct]}}^{(0)})_{ij} \sim \mathcal{N}(0, 1)$. $\mathbf{M}^{(l)} \in \mathbb{R}^{d_{\text{ct}} \times d_{\text{ct}}}$ is a feature transformation matrix for residual connection and $\beta$ is a control factor on compressed token evolution. Notably, setting $\beta > 0$ to evolve the compressed token is crucial for achieving good result, which will be shown in Cor. 2.5 and in experiment 3.2. The function $\mathcal{F}_{\text{comp}} : \{\mathbb{R}^d\}_n (\times \{\mathbb{R}^{d_{\text{ct}}}\}_m) \to \{\mathbb{R}^{d_{\text{ct}}}\}_m$ is a matrix mapping that defines the *compression* of information from main tokens to compressed tokens, giving the *dynamic* of $\mathbf{x}_{\texttt{[ct]}}$. This parameterized compression mapping must be at least as expressive as MLPs and is critical in our latter analysis. We provide instantiation and intuitions regarding how it relates to attention bottleneck in Sec. 2.4. Below, we detail how the $\mathbf{x}_{\texttt{[ct]}}$ is formalized to a regularizer via *information propagation*.

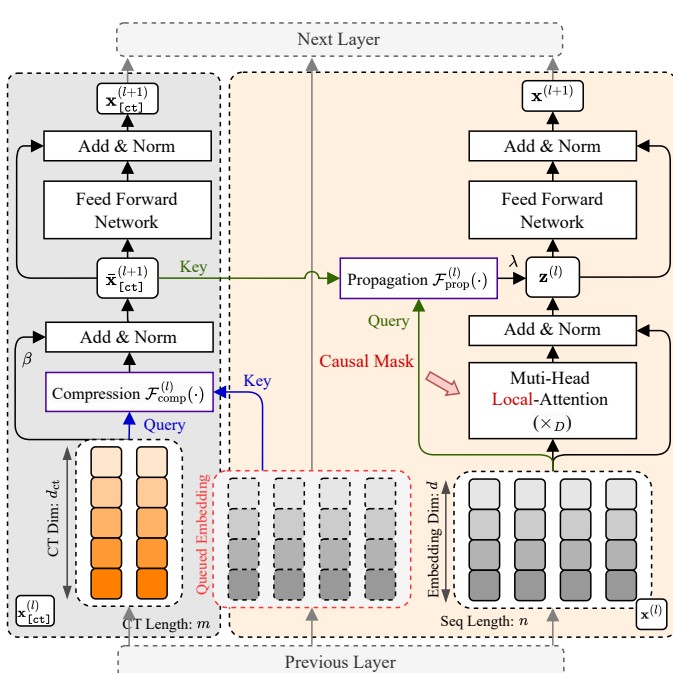

Figure 4: **The BRL-Attention block (enc/dec-only).** The algorithm receives main token $\mathbf{x}$, compressed token $\mathbf{x}_{\texttt{[ct]}}$ for layer $l$, and queued history tokens (on training phase only when causal training is enabled, otherwise the history are the same as main tokens) as input. Within each layer, we first compute the multi-head generalized attention by Eq. (1), under theoretical guarantee Thm. 2.9, the instantiation of $D$ can be quite flexible. In our case, we employ sparse local-attention *e.g.* (Beltagy et al., 2020) as a linear complexity instantiation. Then the context $\mathbf{x}$ in another branch is manipulated with $\mathbf{x}_{\texttt{[ct]}}$ by $\mathcal{F}_{\text{comp}}$ to get the evolved $\mathbf{x}_{\texttt{[ct]}}$. We then construct the regularizer $\mathcal{F}_{\text{prop}}$ by $\mathbf{x}$ and the updated $\mathbf{x}_{\texttt{[ct]}}$. Notably, the $\mathbf{x}$ and $\mathbf{x}_{\texttt{[ct]}}$ are both processed by their respective gated FNN before the next layer. Optionally, the $\mathbf{x}$ in $\mathcal{F}_{\text{prop}}$ could be processed by feature map $\phi/\phi_{\text{spiky}}$ to facilitate a lossless regularizer (see Cor. 2.5). We detail the algorithm in Alg. 1 in Appendix.

[**Practice**] *Regularizer via Compressed Information Propagation*. With the evolution of compressed token defined layer-wise according to Eq. (7), we are ready to instantiate the Bottleneck Regularizer, formulated as a compression operation $\mathcal{F}_{\text{comp}}$ in Eq. (4), defined by

$$\mathbf{z}^Q = \phi(\mathbf{x}^{(l)}) \in \mathbb{R}^{n \times c}, \quad \mathbf{z}^K, \mathbf{z}^V = \mathbf{x}_{\texttt{[ct]}} \mathbf{W}_z^{\{K,V\}} \in \mathbb{R}^{m \times c}, \mathbb{R}^{m \times d}, \tag{8}$$

$$\mathcal{F}_{\text{prop}}^{(l)}(\mathbf{x}^{(l)}; \mathbf{x}_{\texttt{[ct]}}, \lambda) = \lambda \sigma_{\text{attn}} \left( \frac{\mathbf{z}^Q (\mathbf{z}^K)^\top}{\sqrt{d}} \right) \mathbf{z}^V \in \mathbb{R}^{n \times d}. \tag{9}$$

For both encoder-only and causal models, $\mathbf{x}^{(l)}$ is the current input. $\phi : \mathbb{R}^d \rightarrow: \mathbb{R}^c$ is a parameterized feature transformation for aligning feature spaces. Optionally, we could introduce $\phi$ as a spikiness enforcer (Zhang et al., 2024), $\phi_{\text{spiky}}(\mathbf{v}) = [\exp(\mathbf{v}\mathbf{w}_{1\dots c} + \mathbf{b})]$ (which has close representation as Eq. (10) such that theoretically benefits Prop. 2.6). The $\mathcal{F}_{\text{prop}}$, defined by $\mathcal{F}_{\text{prop}} : \{\mathbb{R}^d\}_n \times \{\mathbb{R}^{d_{\text{ct}}}\}_m \rightarrow \{\mathbb{R}^d\}_n$, can be viewed as a cross sequence message passing that pulls the information from $\mathbf{x}_{\texttt{[ct]}}$ back to main tokens. $\mathbf{W}_z^K \in \mathbb{R}^{d_{\text{ct}} \times c}$ and $\mathbf{W}_z^V \in \mathbb{R}^{d_{\text{ct}} \times d}$ are KV weights. The attention function $\sigma_{\text{attn}}(\mathbf{u}) = f_{\text{Laplace/Softmax}}(\mathbf{u} + \mathbf{b}_{\text{rel}})$ can be implemented as Softmax for regular attention, or Squared ReLU/Laplace function (Ma et al., 2022) for better convergence speed and training stability. The $\mathbf{b}_{\text{rel}} \in \mathbb{R}^{n \times m}$ is the relative positional bias, which can be drawn from approaches *e.g.* (Raffel et al., 2020; Su et al., 2024; Ke et al., 2020; Press et al., 2021). We employ $\mathbf{x}_{\texttt{[ct]}} = \mathbf{x}_{\texttt{[ct]}}^{(l+1)}$ for each layer of Eq. (9). We defer the explanation of employing the evolved $\mathbf{x}_{\texttt{[ct]}}$ layer-wise in Cor. 2.5. The overall complexity of Eq. (9) is roughly $\mathcal{O}(mn)$, which scales linearly with sequence length $n$.

[**Theory**] *Propagation Mapping as Regularizer Preserves Comparable Expressibility to Kernelized Attention*. In the following, we theoretically justify that: *Adding regularizer $\mathcal{F}_{\text{prop}}$ to the generalized attention function $\mathcal{F}_{\text{gen}}$ induces no degradation on attention expressibility* despite a positive $\lambda$. Define the *random feature map* (Choromanski et al., 2020) $\phi_{\text{rfm}} : \mathbb{R}^d \rightarrow \mathbb{R}^c$ for $\mathbf{v} \in \mathbb{R}^d$ as

$$\phi_{\text{rfm}}(\mathbf{v}) = \exp(-\|\mathbf{v}\|_2^2/2) \cdot [\exp(\mathbf{v}\mathbf{w}_1), \exp(\mathbf{v}\mathbf{w}_2), \cdots, \exp(\mathbf{v}\mathbf{w}_c)]/\sqrt{c}, \tag{10}$$

where $\mathbf{w}_{1\dots c}$ are random transformations drawn from $\mathcal{N}(0, \mathbf{I}_d)$. According to (Choromanski et al., 2020), $\phi_{\text{rfm}}$ is one of the instantiation of $\phi$ in Eq. (3) that facilitate a softmax kernel-approximation. In the analysis below, we make the following assumptions on Eq. (9): (C1) $\phi(\cdot) \approx \phi_{\text{rfm}}(\cdot)$ and (C2) $d_{\text{ct}} \approx c(1 + d)$. With these assumptions, we arrive at the following proposition:

**Proposition 2.4.** *At each layer $l$, let $\tilde{\mathbf{x}}_{[ct]}^{(l+1)} = [\sum_{j=1}^n \phi_{\text{rfm}}(\mathbf{k}_j^{(l)})^\top \| \text{flatten}(\sum_{j=1}^n \phi_{\text{rfm}}(\mathbf{k}_j^{(l)})^\top \mathbf{v}_j^{(l)})]$, where $\|$ is the concatenation operator. The $\text{flatten}(\cdot)$ operation reshapes input $\mathbb{R}^{c \times d} \rightarrow \mathbb{R}^{cd}$ to raster order. Then, if we* **force** $\mathbf{x}_{[ct]}^{(l+1)} = \tilde{\mathbf{x}}_{[ct]}^{(l+1)}$, *Eq. (9) can sufficiently approximate the kernalized* **self***-attention (i.e., Eq. (3)).*

Recall that $\tilde{\mathbf{x}}_{[ct]}^{(l+1)}$ given in Prop. 2.4 is also a vector matching the shape $\mathbb{R}^{d_{ct}}$ of compressed tokens in Eq. (7) (since $c(1 + d) = d_{ct}$ under (C2)). Therefore, the proposition essentially tells the following Corollary:

**Corollary 2.5.** *With some **particular instantiation** of function $\mathcal{F}_{prop}(\cdot)$ powered by particular choice of $\phi(\cdot)$, there exists some **fixed** compressed token $\mathbf{x}_{[ct]}$ as input to $\mathcal{F}_{prop}$, such that $\mathcal{F}_{prop}(\mathbf{x}; \mathbf{x}_{[ct]})$ is essentially an attention-approximation.*

Cor. 2.5 reflects the benefit of evolving $\mathbf{x}_{[ct]}$ dynamically instead of sharing across layers: *(1)* $\mathbf{x}_{[ct]}^{(l+1)}$ with evolution Eq. (7) is capable on approximating $\tilde{\mathbf{x}}_{[ct]}^{(l+1)}$ of Prop. 2.4 for all $l$; and *(2)* with $\mathbf{x}_{[ct]}$ more expressive than $\tilde{\mathbf{x}}_{ct}^{(l+1)}$, $\mathcal{F}_{prop}$ is at least as expressive as attention approximation. Therefore, adding such particular $\mathcal{F}_{prop}$ under Prop. 2.4 results in no degradation in expressibility. Recall the instantiation of $\mathcal{F}_{prop}$ in Eq. (9). We are then interested in how such a border definition of $\mathcal{F}_{prop}$ could impact the expressibility. Denote $\|\cdot\|$ for vectors as $l_2$ norm and for matrices as spectral norm, we state the following proposition.

**Proposition 2.6.** *Under assumptions that: (A1) Given feature space $\mathcal{X} \subset \mathbb{R}^d$, for all $i \in [1, n]$, the token feature $\mathbf{x}_i$ satisfies $\mathbf{x}_i \in \mathcal{X}$ and $\|\mathbf{x}_i\| \leq r_x$. Similarly, we assume the compressed token $[\mathbf{x}_{[ct]}]_j$ for $j \in [1, m]$ is bounded by $r_{ct}$. (A2) All weight matrices e.g. that feature transformations like attention parameters satisfy $\|\mathbf{W}_*\| \leq r_W$. We have: A parameterized network $f(\cdot)$ that is as expressive as MLP with $\mathcal{O}(1)$ width and depth can approximate $\phi_{rfm}(\mathbf{q}_i)$, $\sum_j \phi_{rfm}(\mathbf{k}_j)$ and $\sum_j \phi_{rfm}(\mathbf{k}_j)^\top \mathbf{v}_j$ arbitrarily well on the compact domain.*

Prop. 2.6 essentially tells that *(1)* $\phi(\cdot)$, if simply defined as $MLP_\phi : \mathbb{R}^d \to \mathbb{R}^c$ for instance, can approximate $\phi_{rfm}(\mathbf{q}_i)$ arbitrarily well (a spiky $\phi = \phi_{spiky}$ facilitate better approximation); and *(2)* in Eq. (7), if disregarding the skip connection *i.e.*, $\beta = 0$, then $\mathcal{F}_{comp}$ with some simple MLP instantiation (*e.g.*, $\sum_j MLP_\mathcal{F}(\mathbf{x}_j)$ where $MLP_\mathcal{F} : \mathbb{R}^{d_{ct}} \to \mathbb{R}^{d_{ct}}$), can approximate $\tilde{\mathbf{x}}_{[ct]}^{(l+1)}$ arbitrarily well. Therefore, if we can essentially approximate both $\phi_{rfm}(\mathbf{q}_i)$ and $\tilde{\mathbf{x}}_{[ct]}^{(l+1)}$, then by Prop. 2.4, the propagation mapping $\mathcal{F}_{prop}$, also defined as a function of $\phi_{rfm}(\mathbf{q}_i)$ and $\tilde{\mathbf{x}}_{[ct]}^{(l+1)}$, can approximate the kernelized form of self-attention (Eq. (3)) arbitrarily well[1]. Finally, the broad definition of $\mathcal{F}_{prop}$ as in Eq. (9) could also inherit the benefits instructed after Cor. 2.5. In the following, we discuss how appropriate instantiation of the compression mapping $\mathcal{F}_{comp}$ impacts the attention bottleneck bound.

## 2.4 The Information Compression and Sensitivity Analysis of BRL-Attention

This section tackles the RQ: *Given a squashed pattern $D$, how a concrete compression mapping lets the $\mathcal{F}_{gen}$ approximate full attention yet keep the linear complexity and offers extended sensitivity bound.* We investigate this by instantiating $\mathcal{F}_{comp}$ as a differential cross-attention that uses rescaled dual score matrices. This construction yields the wider bounded-error Jacobian and preserves linear complexity.

**[Practice]** *Information Compression*. We now introduce an instantiation of $\mathcal{F}_{comp}(\mathbf{x}_{[ct]}; \mathbf{x})$ in Eq. (7). Let $\bar{\mathbf{x}}$ be the history of $\mathbf{x}$ on time $t$ of sequence modeling. Define

$$\mathbf{h}^Q = \mathbf{x}_{[ct]}\mathbf{W}_h^Q \in \mathbb{R}^{m \times d_{ct}}, \quad \mathbf{h}^{K1}, \mathbf{h}^{K2}, \mathbf{h}^V = \bar{\mathbf{x}}\mathbf{W}_h^{\{K1,K2,V\}} \in \mathbb{R}^{n \times d_{ct}}, \tag{11}$$

$$\mathcal{F}_{comp}(\mathbf{x}_{[ct]}; \mathbf{x}) = (\mathbf{S}^{K1} - \gamma\mathbf{S}^{K2})\mathbf{h}^V \in \mathbb{R}^{m \times d_{ct}}, \text{ where } \mathbf{S}^{K1}, \mathbf{S}^{K2} = \sigma_{attn}(\frac{\mathbf{h}^Q(\mathbf{h}^{\{K1,K2\}})^\top}{m\sqrt{d_{ct}}}) \in \mathbb{R}^{m \times n}. \tag{12}$$

Notably, for self-attention, $\bar{\mathbf{x}}$ is identical to $\mathbf{x}$, while for autoregressive training, $\bar{\mathbf{x}}$ is the last history $\mathbf{x}$ of the same block size (more details will be illustrated in Appendix B.2). Compared to the formulation of $\mathcal{F}_{prop}(\mathbf{a}, \mathbf{b})$ in Eq. (9), we regard $\mathbf{x}_{[ct]}$ as the query and construct keys and values with main tokens $\mathbf{x}$. The $\sigma_{attn}$ now have relative bias $\mathbf{b}_{rel} \in \mathbb{R}^{m \times n}$. In addition, we make the following modifications: *(M1)* The attention scaler is changed from $\sqrt{d}$ to $m\sqrt{d_{ct}}$. *(M2)* Inspired by differential transformer (Ye et al., 2024), we construct two attention scores $\mathbf{S}^{K1}$ and $\mathbf{S}^{K2}$ with same $\mathbf{x}_{[ct]}$ as query and different transformations of $\mathbf{x}$ as keys. We regard their re-scaled subtraction as the final attention score.

**[Theory]** *Alleviated Attention Bottleneck (Information Squashing) via Compression Map*. The formulation as Eq. (12) offers the following properties:

---

[1]Remarkably, the error gap between kernelized and full attention $\|\phi(\mathbf{q}_i/\sqrt{\tau})\phi(\mathbf{k}_j/\sqrt{\tau})^\top - \kappa(\mathbf{q}_i/\sqrt{\tau}, \mathbf{k}_j/\sqrt{\tau})\|$ is bounded by $\mathcal{O}(\sqrt{\exp(6r_x r_W/\tau)/m\epsilon})$ according to (Thm. 1, (Wu et al., 2022b))

**Proposition 2.7.** *Assume for simplicity in Eq. (7) that $\beta = 0$, and $\mathcal{F}_{\text{comp}}$ is defined according to Eq. (12). Then in terms of the sensitivity bound of $\mathcal{F}_{\text{prop}}$, the modification M1 resolves the bound scaling with $\mathbf{x}_{[ct]}$ sequence length $m$, and M2 provides a controlled bound with factor $\gamma$ compared to vanilla cross-attention. The sensitivity bound will be*

$$\left\| \frac{\partial \mathcal{F}_{\text{prop}}^{(l)}(\mathbf{x}^{(l)}; \mathbf{x}_{[ct]}^{(l+1)})_i}{\partial \mathbf{x}_p^{(l)}} \right\| \leq \mathcal{O}\left( \frac{r_x r_{\text{ct}} r_W^6}{\sqrt{dd_{\text{ct}}}} (1 + |\gamma|) \right), \tag{13}$$

*compared to the bound of $\mathcal{O}(\frac{r_x r_{\text{ct}} r_W^6}{\sqrt{dd_{\text{ct}}}})$ without M2. This bound is applicable to $l \in [0, L]$, which helps the alleviation of attention bottleneck compared to with $\mathcal{F}_{\text{gen}}$ only (as shown in Prop. 2.2, the bound is consistently zero when $M$ is large). For details w.r.t. the lower-bound, we refer the readers to Appendix E.7.*

*Remark* 2.8. Augmenting the compression $\mathcal{F}_{\text{comp}}$ as Eq. (12) to differential form (according to M2) could also potentially improve the robustness of learning useful information to those irrelevant, which benefits the retrieval of information in a long noisy sequence. While augmenting the propagation mapping $\mathcal{F}_{\text{prop}}$ does not offer such a benefit for a row-wise $\sigma_{\text{attn}}$, we keep it in regular cross-attention form (however, one may still augment $\mathcal{F}_{\text{prop}}$ to differential form if column-wise normalization is favored). We justify in detail regarding the above phenomenon via the concept of Signal-to-Noise Ratio (SNR) in Remark D.3.

Consider the BRL-Attention layer in the form $\mathcal{F}_{\text{BRL}}^{(l)}(\mathbf{x}^{(l)})_i = \mathcal{F}_{\text{gen}}^{(l)}(\mathbf{x}^{(l)}; D_{\text{sw}})_i + \mathcal{F}_{\text{prop}}^{(l)}(\mathbf{x}^{(l)}, \mathbf{x}_{[ct]}^{(l+1)})_i$, which is equivalent to initializing Eq. (4) by $D = D_{\text{sw}}$ and $\lambda = \{\mathbf{1}\}_H$. Recalling Prop. 2.2, with the result of Ineq. (6), we have $\forall l < \lceil \frac{2M}{w-1} \rceil - 1 : \| \frac{\partial \mathcal{F}_{\text{gen}}^{(l)}(\cdot; D_{\text{sw}})_i}{\partial \mathbf{x}_p^{(l)}} \| = \| \frac{\partial \mathcal{F}_{\text{gen}}^{(l)}(\cdot; D_{\text{sw}})_i}{\partial \mathbf{x}_p^{(0)}} \| = 0$. This implies that the information of token $p$ cannot be captured by token $i$ within $\lceil \frac{2M}{w-1} \rceil - 1$ layers of attention propagation. When adding the regularizer $\mathcal{F}_{\text{prop}} \circ \mathcal{F}_{\text{comp}}$ to $\mathcal{F}_{\text{gen}}$, we essentially extend the sensitivity bound to non-zero, which implies $\mathcal{F}_{\text{BRL}}$ of token $i$ is more capable of receiving (potentially bottlenecked-)information from $p$-th token from previous layers, which alleviates the squashness of attention propagation.

**[Theory]** *Carefully Tailored Compression-Propagation Offers Comparable Expressibility to Full-Attention*. To show that $\mathcal{F}_{\text{prop}} \circ \mathcal{F}_{\text{comp}}$ does not degrade the performance, finally, we state Thm. 2.9 to show that the regularized generalized attention $\mathcal{F}_{\text{gen}} + \mathcal{F}_{\text{prop}} \circ \mathcal{F}_{\text{comp}}$, with compression-propagation formulated as Eqs. (9, 12), is as expressive as full-attention on general long sequence tasks.

**Theorem 2.9** (Expressibility of $\mathcal{F}_{\text{BRL}}$). *Given $1 \leq p < \infty$ and $\epsilon > 0$, for any continuous functions $\mathcal{F}_{\text{con}} : [0, 1]^{n \times d} \to \mathbb{R}^{n \times d}$, let the attention function $\sigma_{\text{attn}}$ of $\mathcal{F}_{\text{prop}}$ be column-wise softmax. Then let $d_p$ be the $l_p$ distance, there exists a BRL-Former with **arbitrary** sparse-attention $D$ such that*

$$d_p(\mathcal{F}_{\text{con}}, \mathcal{F}_{\text{BRL}}) = d_p(\mathcal{F}_{\text{con}}, \mathcal{F}_{\text{gen}}(\cdot; D) + \mathcal{F}_{\text{prop}}(\cdot; \mathcal{F}_{\text{comp}}(\cdot))) \leq \epsilon. \tag{14}$$

*Sketch Proof.* *(1)* We first show Prop. E.2 where $\mathcal{F}_{\text{BRL}} = \mathcal{F}_{\text{gen}}(\cdot; D) + \mathcal{F}_{\text{prop}}(\cdot; \mathcal{F}_{\text{comp}}(\cdot))$ can simulate generalized attention with a star graph $S$ (given as Def. E.1), $\mathcal{F}_{\text{gen}}(\mathbf{x}; D \cup S)$, with any sparse pattern $D$ where $D \cap S = \emptyset$. This means $\mathcal{F}_{\text{BRL}} \approx \mathcal{F}_{\text{gen}}(\mathbf{x}; D \cup S)$. *(2)* We employ Thm. E.3 (Zaheer et al., 2020) that states: for any continuous function $\mathcal{F}_{\text{con}}$ and $\epsilon > 0$, there exists a sparse Transformer $\mathcal{F}_{\text{gen}}(\cdot; D \cup S)$ such that $d_p(\mathcal{F}_{\text{con}}, \mathcal{F}_{\text{gen}}(\cdot; D \cup S)) \leq \epsilon$. *(3)* Since $\mathcal{F}_{\text{BRL}}$ can approximate $\mathcal{F}_{\text{gen}}(\cdot; D \cup S)$, and $\mathcal{F}_{\text{gen}}(\cdot; D \cup S)$ can approximate any $\mathcal{F}_{\text{con}}$, it follows that $d_p(\mathcal{F}_{\text{con}}, \mathcal{F}_{\text{BRL}}) \leq \epsilon$ with an arbitrary sparse pattern $D$.

## 2.5 Implementation

We illustrate the framework (self-attention version) of our proposed attention block in Fig. 4. Meanwhile, we detail the batched algorithm of Eq. (4) in Alg. 1 in Appendix. The local-attention part (the $\mathcal{F}_{\text{gen}}$ function in Alg. 1) is implemented based on Phil Wang's implementation.[2]

**Initialization of Parameters.** We allow for a flexible number of compressed tokens ($m$ where $m \geq 1$). Typically, an $m \leq 256$ is good enough for all tasks in our evaluations. We set $c = d_{\text{ct}} = d$ in Eq. (9 for simplicity. For hyperparameter setups: We empirically find setting $\beta = 0.5$ performs well generally. We let $\gamma = \exp(\gamma_Q \gamma_{K1}) - \exp(\gamma_Q \gamma_{K2})$ where default values for the **learnable** $\gamma_{Q/K1/K2}$ are

---

[2]Local-Attention github repo: `https://github.com/lucidrains/local-attention`

`torch.normal(H,mean=0,std=0.1)`. Practically, it is good to initial mean of learnable $\gamma$ to 0 (similar to (Zhang et al., 2023)) to facilitate a *cold start*, which does not affect the embedding quality if the differential form offers negative impact. Additionally, we observed in Fig. 7(b) that setting disabling the learnability of $\gamma$ offers negative impact. Finally, we select the regularizer weight $\lambda$ in Eq. (4, 7) from $0.1 \sim 1.0$`*torch.ones(H)` where we empirically find $\leq 0.5$ generally works well. Still notably, as $\lambda$ is also learnable, the choice of which is not sensitive to the final result (which can be observed in Fig. 7(a)).

**Computational Complexity Analysis.** Deferred to Appendix B.

## 3 Experiments

### 3.1 Time and Memory Complexity Evaluation

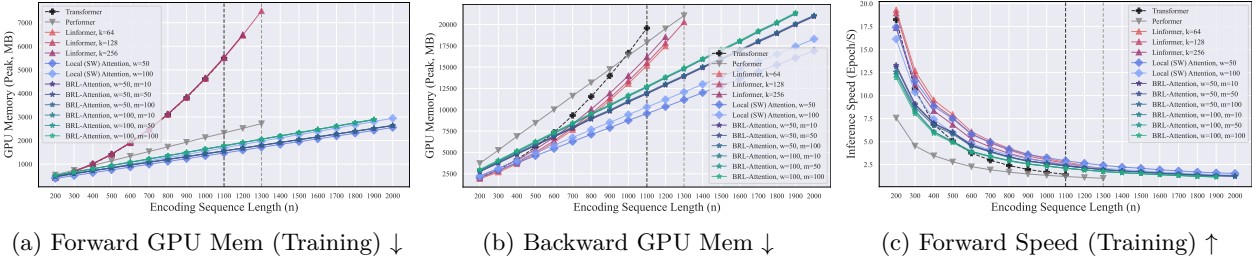

(a) Forward GPU Mem (Training) ↓    (b) Backward GPU Mem ↓    (c) Forward Speed (Training) ↑

Figure 5: Comparison of the computational requirements for a Forward/Backward pass for **Encoder-Decoder** based models (batch size 16, 1-layer of encoder and 8-layers of decoder with 8 heads and 512 hidden dims) on simple copy task. Dotted lines denote *out of memory* of corresponding models. Our BRL-Attention scales linearly with the sequence length, unlike Full-Attention (Softmax), which scales with the square of the sequence length both in memory and time.

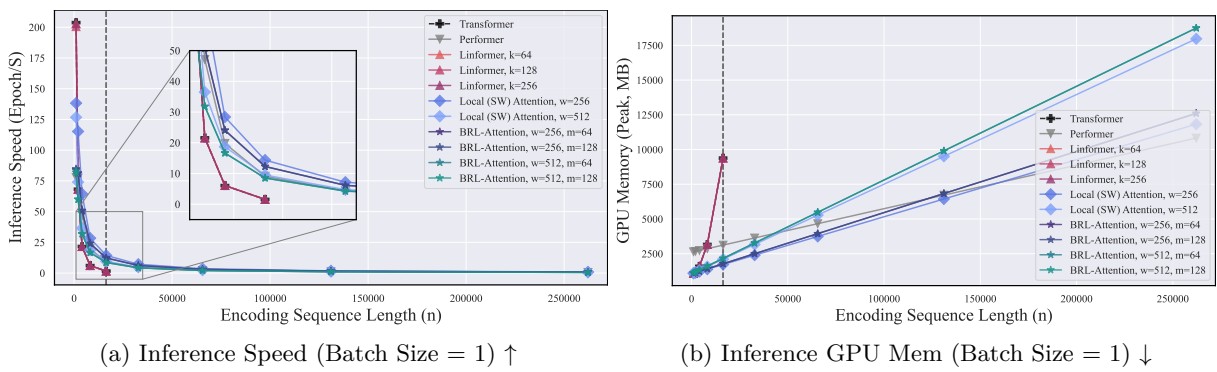

(a) Inference Speed (Batch Size = 1) ↑    (b) Inference GPU Mem (Batch Size = 1) ↓

Figure 6: Comparison of inference efficiency of various **Encoder-Only** transformers (single batch, 4-layers of transformers, 4 heads and 256 hidden dims). Our model achieves $\sim 13\times$ (further evaluation leads to the OOM of Full-Attention) of inference speed on a long sequence while scaling linearly with sequence length.

We evaluate transformers with various attentions on *Autoregressive Encoder-Decoder Copy Task* – a sequence modeling experiment in which an encoder first processes a given input sequence, and then an autoregressive decoder is trained to replicate that same sequence as its output. Notably, all baselines are capable in convergence, and we only employ it as a debugging baseline for evaluating the memory and time consumption for encoder-decoder based models training. Specifically, we evaluate Full-Attention, Performer, Linformer ($k \in \{64, 128, 256\}$), Local-Attention ($w \in \{50, 100\}$) and our BRL-Attention ($w \in \{50, 100\}, m \in \{10, 50, 100\}$). The experiment ran under batch size 16, with a 1-layer of encoder, 8-layer of decoder, 8 heads, and 512 hidden dims. We benchmark with sequence length $n \in [200, 2000]$ with

step size 100. As demonstrated in Fig. 5, the Full-Attention Transformer, Performer, and Linformer with all settings all suffer from out-of-memory when sequence length exceeds $1100 \sim 1300$. Our BRL-Attention, for the same window size $w$ as Local-Attention, the BRL regularization only adds negligible computation, which also results in a theoretically $\mathcal{O}(n)$ memory complexity according to Sec. 2.5. Verified in Fig. 5, similar to Local-Attention, the BRL-Attention scales linearly with the sequence length, unlike Full-Attention (Softmax), which scales with the square of the sequence length both in memory and time.

Additionally, we benchmark the inference-only speed and memory performance with a single batch. In this experiment, we evaluate Full-Attention, Performer, Linformer ($k \in \{64, 128, 256\}$), Local-Attention ($w \in \{256, 512\}$) and our BRL-Attention ($w \in \{256, 512\}, m \in \{64, 128\}$). All the architectures are encoder/decoder only, with only self-attention. We test 4-layers of enc/dec, 4 heads, and 256 hidden dims. As no backward is required, we can scale $n$ up to $2^{18}$ length with a single 24GB RTX-4090 GPU. In essence, we range $n$ in $\{2^{10}, 2^{11}, ..., 2^{18}\}$. We demonstrate the inference speed and memory in Fig. 6. The Full-Attention, similar to Linformer, gives out-of-memory on $n > 2^{14}$, which is incapable of very long sequence inference. The Performer, different from its autoregressive training performance (worse than BRL-Attention on both time and memory cost when scaling to longer sequences), performs well on inference-speed. Our models achieve $> 13\times$ of inference speed on sequence length $n \geq 2^{14}$ compared to the Full-Attention counterpart. This verifies the efficiency of our method.

Table 1: **Results on Long Range Arena Benchmark.** We compare our method to three major classes of efficient transformers, namely, the *full attention*, *low-rank kernel*, and *pattern-based*, as mentioned in the preliminaries. The best results are in boldface, and the second bests are underlined. $X$-marks in Path-X denote chance accuracy.

| Class | Methods | Linear | ListOps ↑ (2k) | Text ↑ (4k) | Retrieval ↑ (8k) | Image ↑ (1k) | Pathfinder ↑ (1k) | Path-X ↑ (16k) | Average ↑ – |
|---|---|---|---|---|---|---|---|---|---|
| Full-Attention | Transformer | ✗ | 36.37 | 64.27 | 57.46 | 42.44 | 71.40 | ✗ | 54.39 |
| | Transformer (our-imp) | ✗ | 47.90 | 79.08 | 82.31 | 75.04 | 76.64 | 84.72 | 72.19 |
| Low-Rank Kernels | Linformer (Wang et al., 2020) | ✓ | 35.70 | 53.94 | 52.27 | 38.56 | 76.34 | ✗ | 51.36 |
| | Linear Trans (Katharopoulos et al., 2020) | ✓ | 16.13 | 65.90 | 53.09 | 42.34 | 75.30 | ✗ | 50.55 |
| | Performer (Choromanski et al., 2020) | ✓ | 18.01 | 65.40 | 53.82 | 42.77 | 77.05 | ✗ | 51.41 |
| | Luna (Ma et al., 2021) | ✓ | 35.33 | 65.11 | 59.61 | 38.67 | 77.80 | ✗ | 55.30 |
| | cosFormer (Qin et al., 2022b) | ✓ | 37.90 | 63.41 | 61.36 | 43.17 | 70.33 | ✗ | 55.23 |
| | Flowformer (Wu et al., 2022a) | ✓ | 38.70 | 64.29 | 62.24 | 43.20 | 73.95 | ✗ | 56.48 |
| Learnable / Fixed Patterns | Local-Attn | ✓ | 15.82 | 52.98 | 53.39 | 41.46 | 66.63 | ✗ | 46.06 |
| | Sparse Trans (Child et al., 2019) | ✗ | 17.07 | 63.58 | 59.59 | 44.24 | 71.71 | ✗ | 51.24 |
| | Longformer (Beltagy et al., 2020) | ✓ | 35.63 | 62.85 | 56.89 | 42.22 | 69.71 | ✗ | 53.46 |
| | BigBird (Zaheer et al., 2020) | ✓ | 36.05 | 64.02 | 59.29 | 40.83 | 74.87 | ✗ | 55.01 |
| | Sliceformer (Yuan & Xu, 2023) | ✗ | 37.65 | 64.60 | 62.23 | 48.02 | 82.04 | ✗ | 58.91 |
| | Reformer (Kitaev et al., 2020) | ✗ | 37.27 | 56.10 | 53.40 | 38.07 | 68.50 | ✗ | 50.67 |
| | Sinkhorn Trans (Tay et al., 2020a) | ✗ | 33.67 | 61.20 | 53.83 | 41.23 | 67.45 | ✗ | 51.39 |
| | Synthesizer (Tay et al., 2021) | ✗ | 36.99 | 61.68 | 54.67 | 41.61 | 69.45 | ✗ | 52.88 |
| Fixed Patterns + Low-Rank Regularizer | BRL-Former ($m = 64$) | ✓ | 47.37 | 80.29 | 82.69 | 75.75 | 76.94 | 85.26 | 74.72 |
| | BRL-Former ($m = 128$) | ✓ | 49.14 | 80.33 | 82.98 | **76.47** | 76.06 | 86.16 | 75.19 |
| | BRL-Former ($m = 256$) | ✓ | **49.98** | **80.90** | **83.22** | 76.20 | **77.48** | **86.89** | **75.78** |

## 3.2 Experiments on Sequence Modeling

**Encoder-Only Sequence Modeling.** We evaluate BRL-Former on the Long Range Arena (LRA) (Tay et al., 2020b), and compare against established baselines in both full-attention and efficient-attention categories. *(1) Full-Attention.* We regard the standard Transformer as the baseline, which has quadratic complexity in sequence length. The 'our-imp' denotes our implementation with RoPE (Su et al., 2024) based on (Amos et al., 2023) *(2) Low-Rank & Kernel Approximation*, which reduces complexity by projecting queries/keys or by approximating the attention matrix with random features. *Learnable/Fixed Attention Patterns*, which leverages windowed or dilated sparse patterns to reduce attention complexity from $\mathcal{O}(n^2)$ to approximately $\mathcal{O}(n) \sim \mathcal{O}(n \log n)$. In our experiments, we employ the LRA training procedure as instructed in (Amos et al., 2023) as baselines for comparison.

Main hyperparameters: (1) The window size $w$ of the local pattern $D$, which scales the local receptive field. (2) The compressed token length $m$ that aids in mitigating the attention *over-squashing*. We explore three settings of ($m = w \in \{64, 128, 256\}$) alongside $w = 512$ for all local-attention-based models (*e.g.*

Local-Attn, Longformer). The hyperparameter settings for each subtask for the original-implementation and our-implementation are delegated to Tab. 11 and Tab. 12, respectively.

Tab. 1 reports the final LRA scores. As shown, the BRL-attention with $m \geq 128$ achieves either the best or runner-up performance across all LRA subtasks, which validates our capability to encode long sequences. In Tab. 2, we additionally compare the performance of BRL-attention to the variant with $\gamma = 0$ (disabling the Attn-Diff). We observe that the Attn-Diff contributes positively to the final performance, which is also validated in the following autoregressive sequence modeling experiments.

Table 2: Ablation study on LRA *w.r.t* the effect of Attention Differentiation.

| Model | ListOps ↑ | Text ↑ | Retrieval ↑ | Image ↑ | Pathfinder ↑ |
|---|---|---|---|---|---|
| Transformer | 36.37 | 64.27 | 57.46 | 42.44 | 71.40 |
| Transformer (our-imp) | 47.90 | 79.08 | 82.31 | 75.04 | 76.64 |
| Local-Attn (m=0) | 15.82 | 52.98 | 53.39 | 41.46 | 66.63 |
| BRL-Former (w/o Attn-Diff) | 48.10 | 78.79 | 81.95 | 74.53 | 76.70 |
| BRL-Former | **49.98** | **80.90** | **83.22** | **76.20** | **77.48** |

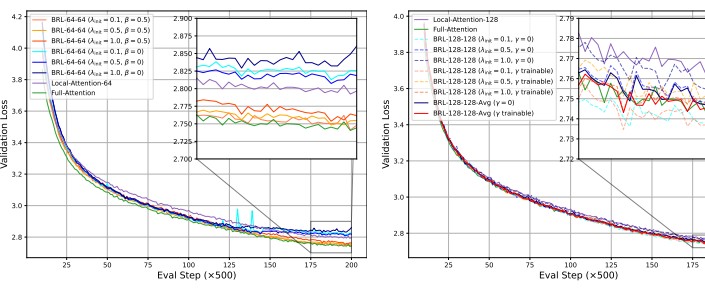

(a) WikiText-103 validation loss over steps. The window size and compression length are both 64.

(b) WikiText-103 validation loss over steps. The window size and compression length are both 128.

Figure 7: Losses under various setups of BRL-Attention.

Table 3: **Results on WikiText-103.** The best/second are bold/underlined. (our-imp) denotes our implementation with customized model configurations based on `nanoGPT`, fewer parameters than (Qin et al., 2022a).

| Method | PPL (val) ↓ | PPL (test) ↓ |
|---|---|---|
| Transformer | 29.63 | 31.01 |
| Transformer-LS (Zhu et al., 2021) | 32.37 | 32.59 |
| FLASH (Hua et al., 2022) | 33.18 | 34.63 |
| Linear Trans (Katharopoulos et al., 2020) | 32.63 | 34.25 |
| Performer (Choromanski et al., 2020) | 75.29 | 77.65 |
| TransNormer (Qin et al., 2022a) | 29.57 | 31.01 |
| Transformer (our-imp) | 21.31 | **22.03** |
| Local-Attention (our-imp) | 22.08 | 23.74 |
| BRL-Former | **20.56** | 22.11 |

**Decoder-Only Autoregressive Sequence Modeling.** We study the autoregressive language modeling on WikiText-103 (Merity et al., 2016). We detail the model parameter configuration in Tab. 13. We defer the study with OpenWebText (Gokaslan et al., 2019) to Appendix A.3.

In Fig. 7, we compare the performance of BRL-attention to local/full-attention under various settings. With various initial $\lambda$ values, which is a coefficient of propagation mapping $\mathcal{F}_{\text{prop}}$, in (a), we additionally study the impact of $\beta$ in Eq. (7), where $\beta = 0$ indicates no residual connection on the evolution of $\mathbf{x}_{[\text{ct}]}$. We observe that with only $m = w = 64$, the BRL-attention significantly surpasses the local attention, which indicates the attention bottleneck is relieved. Different setups of $\lambda_{\text{init}}$ do not heavily impact the final result as they are trainable. However, a good $\lambda_{\text{init}}$ is around $0.1 \sim 0.5$. In (b), we additionally study the impact of $\gamma$ in compression mapping $\mathcal{F}_{\text{comp}}$. With $m = w = 128$ ($\ll$ block size 512), the BRL-attention can achieve comparable and better results against full-attention. Nonetheless, as setting $\gamma$ as trainable indicates enabling Attn-Diff, we observe that Attn-Diff benefits the optimization, which leads to lower losses compared to those with $\gamma = 0$. Using perplexity (PPL) as the evaluation metric, the final results are reported in Tab. 3, where baseline results are partially derived from (Qin et al., 2022a) for reference. The BRL-Attention obtains comparable or better perplexity to the vanilla attention and outperforms all existing linear models with a clear margin. Compared to linear methods, BRL-Former achieves substantially lower perplexity, demonstrating the effectiveness of our method in causal models.

## 3.3 Experiments on Large Graph Modeling

We evaluate BRL-Former on node classification tasks using DBLP, ACM, IMDB, and Freebase datasets from the HGB benchmark. DBLP, ACM, and IMDB follow HGB (Lv et al., 2021) guidelines, while Freebase uses the split from (Mao et al., 2023). Dataset details are in Tab. 10. Evaluation metrics include micro/macro-F1.

Baseline models span four categories: (1) Simple MPNNs (GCN, GAT); (2) Message-passing heterogeneous GNNs (RGCN, HAN, HetGNN, Simple-HGN); (3) Transformer-based models (GTN, HGT, NodeFormer, HINormer); and (4) Pure transformers with Poly-Token, evaluated with Full-Attention (PHGT), Local-

Table 4: **Results on heterogeneous node classification datasets.** Vacant positions ($\mathcal{X}$) indicate the models run OOM on the corresponding datasets. We report the average results in 3 runs.

| Methods | DBLP | | IMDB | | ACM | | Freebase | |
|---|---|---|---|---|---|---|---|---|
| | Micro-F1 ↑ | Macro-F1 ↑ | Micro-F1 ↑ | Macro-F1 ↑ | Micro-F1 ↑ | Macro-F1 ↑ | Micro-F1 ↑ | Macro-F1 ↑ |
| GCN (Kipf & Welling, 2016) | 91.47±0.34 | 90.84±0.32 | 64.82±0.64 | 57.88±1.18 | 92.12±0.23 | 92.17±0.24 | 60.23±0.92 | 27.84±3.13 |
| GAT (Veličković et al., 2017) | 93.39±0.30 | 93.83±0.27 | 64.84±0.43 | 58.94±1.35 | 92.19±0.39 | 92.26±0.94 | 65.26±0.80 | 40.74±2.58 |
| RGCN (Schlichtkrull et al., 2018) | 92.07±0.50 | 91.52±0.50 | 62.05±0.15 | 58.85±0.26 | 91.41±0.75 | 91.55±0.74 | 60.82±1.42 | 59.08±1.44 |
| HAN (Wang et al., 2019) | 92.05±0.62 | 91.67±0.49 | 64.63±0.58 | 57.74±0.96 | 90.79±0.43 | 90.89±0.43 | 61.42±3.56 | 57.05±2.06 |
| HetGNN (Zhang et al., 2019) | 92.33±0.41 | 91.76±0.43 | 51.16±0.65 | 48.25±0.67 | 86.05±0.25 | 85.91±0.25 | ✗ | ✗ |
| Simple-HGN (Lv et al., 2021) | 94.46±0.22 | 94.01±0.24 | 67.36±0.57 | 63.53±1.36 | 93.35±0.45 | 93.42±0.44 | 67.49±0.97 | 62.49±1.69 |
| GTN (Yun et al., 2019) | 93.97±0.54 | 93.52±0.55 | 65.14±0.45 | 60.47±0.98 | 91.20±0.71 | 91.31±0.70 | ✗ | ✗ |
| HGT (Hu et al., 2020) | 93.49±0.25 | 93.01±0.23 | 67.20±0.57 | 63.00±1.19 | 91.00±0.76 | 91.12±0.76 | 66.43±1.88 | 60.03±2.21 |
| NodeFormer (Wu et al., 2022b) | 93.68±0.42 | 93.05±0.38 | 65.86±0.42 | 62.15±0.77 | 91.89±0.31 | 92.72±0.84 | 67.01±0.52 | 60.83±1.41 |
| HINormer (Mao et al., 2023) | 94.94±0.21 | 94.57±0.23 | 67.83±0.34 | 64.65±0.53 | 93.15±0.36 | 93.28±0.43 | 67.78±0.39 | **62.67±1.10** |
| Full-Transformer (Lu et al., 2024) | 95.33±0.18 | 94.96±0.17 | 68.81±0.08 | 65.91±0.30 | 93.72±0.40 | 93.79±0.39 | 68.74±1.42 | 61.73±1.86 |
| Local-Attn | 94.96±0.24 | 94.87±0.35 | 67.93±0.14 | 65.45±0.32 | 93.33±0.30 | 93.58±0.24 | 67.78±0.53 | 60.98±0.94 |
| BRL-Former | **95.67±0.20** | **95.35±0.18** | **68.99±0.12** | **66.29±0.46** | **93.78±0.21** | **93.81±0.25** | **69.54±1.06** | 61.80±2.40 |

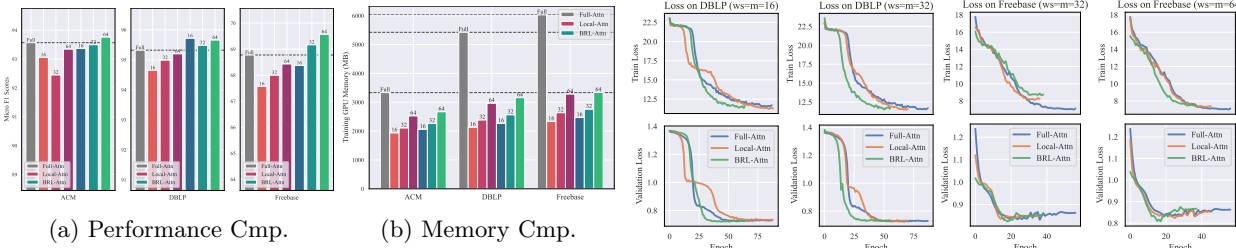

(a) Performance Cmp.      (b) Memory Cmp.

Figure 8: Performance and mem-cost comparison of Transformers with different attention backbones on various setups. We observe that BRL-Attn is superior in performance and memory efficiency (neglectfully more costly than Local-Attn) across all datasets.

Figure 9: Convergence of training/validation losses on heterogeneous graph datasets. We compare Transformer with Full/Local/BRL-Attention. We observe that BRL-Attn typically converges faster than the others on their respective minimum.

Attention, and our BRL-Attention. Baseline results from HGB (Lv et al., 2021) are quoted directly; others are re-evaluated via OpenHGNN.

Tab. 4 presents the node classification results. Our proposed BRL-Former with Poly-Token consistently demonstrates superior performance, outperforming other baselines, including the Full-Attention Transformer, across most scenarios. For the Freebase dataset, (Lu et al., 2024) notes that local structures are particularly significant, as Freebase is a knowledge graph composed of individual facts or triples. BRL-Attention naturally restricts the receptive field to local subgraphs, providing notable advantages on this dataset, surpassing the current state-of-the-art results. Moreover, BRL-Attention outperforms message-passing (H)GNN baselines in nearly all cases, indicating that compressed token propagation effectively resolves the receptive field limitations inherent in Local-Attention and enhances model performance on heterogeneous graphs. Additional analyses in Fig. 8-9 reveal BRL-Attention is superior in performance, being memory efficient (neglectfully more costly than Local-Attn) while consistently converges faster across all datasets to lower loss values, confirming its efficiency and effectiveness on large graphs modeling.

## 4 Conclusion

We presented *Bottleneck-Regularized Linear Attention*, a mechanism that augments sparse/pattern-based attention with a small set of *compressed tokens* to capture long-range dependencies at linear cost. Our theoretical analysis and extensive experiments on sequence and graph benchmarks show that BRL-Attention consistently matches or outperforms full-attention baselines, while being more efficient. Unlike kernel-based methods, BRL-Attention avoids challenging kernel approximations and supports parallel training for both encoders and autoregressive decoders. In essence, the compressed tokens serve as a global reservoir that mitigates over-squashing without requiring costly quadratic attention or specialized global tokens that break causality. These findings position BRL-Attention as an efficient and scalable alternative for regular attention.

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

# A    Additional Evaluations

## A.1    Experiments on Text and Vision Datasets

We evaluate our model on two datasets without graph structure: 20News-Groups (Pedregosa et al., 2011) and Mini-ImageNet (Vinyals et al., 2016). The 20News dataset is a collection of approximately 20,000 newsgroup documents (nodes), partitioned (nearly) evenly across 20 different newsgroups. We take 10 classes from 20 newsgroups and use words (TF-IDF) with a frequency of more than 5% as features. The Mini-ImageNet dataset consists of $84{\times}84$ RGB images from 100 different classes with 600 samples per class. For our experiment use, we choose 30 classes from the dataset, each with 600 images (nodes) that have 128 features extracted by CNN. Since there is no input graph, we use k-NN (over input node features) for artificially constructing a graph for enabling GNN's message passing and the graph-based component. We report the results under the **best** $k \in [5, 20]$ setup for each GNN baseline. A summary of the statistics of each dataset is provided in Tab. 9.

Table 5: **Quantitative results on semi-supervised classification** with Mini-ImageNet and 20News-Groups. We use $k$-NN (with different $k$s) for artificially constructing an input graph. The best and second-best results are highlighted in bold and underlined, respectively, where models with and without graph are compared separately.

| Class | Methods | 20News-Group | Mini-ImageNet |
|---|---|---|---|
| Graph-based ($k$NN $k \in [5, 20]$) | GCN | 65.98±0.68 | 85.96±0.66 |
| | GAT | 64.06±0.44 | 85.41±0.43 |
| | DropEdge | 64.46±0.43 | 85.81±0.65 |
| | IDGL | 65.09±1.23 | 85.66±0.42 |
| | LDS-GNN | **66.15±0.36** | OOM |
| NodeFormer Framework (w/o graph) | Gumbel-Softmax $+ \mathcal{L}_e$ | 64.71±1.33 | 87.45±0.55 |
| | Full-Attn | 64.94±0.16 | 87.46±0.54 |
| | Local-Attn-$m = 2^3$ | 64.54±0.23 | 86.62±0.91 |
| | Local-Attn-$m = 2^4$ | 64.38±0.43 | 87.03±0.52 |
| | Local-Attn-$m = 2^5$ | 64.61±0.57 | 87.17±0.49 |
| | BRL-Attn-$m = w = 2^3$ | 64.81±0.35 | 87.37±0.60 |
| | BRL-Attn-$m = w = 2^4$ | 65.02±0.46 | 87.46±0.63 |
| | BRL-Attn-$m = w = 2^5$ | 65.19±0.69 | **87.55±0.54** |

As depicted in Tab. 5, the NodeFormer with BRL-Attention (without input graph and edge loss $\mathcal{L}_e$) achieves competitive performance against its opponents, including GNN-based baselines (Kipf & Welling, 2016; Veličković et al., 2017; Rong et al., 2019; Chen et al., 2020; Franceschi et al., 2019) and NodeFormer (Wu et al., 2022b) with Gumbel-Softmax/Full/Local attention. For the smaller 20News, our method achieves the second best and outperforms the full-attention method, while the local-attention fails to achieve decent performance. Notably, the BRL-Attention offers 4× of GPU memory reduction compared to LDS on 20News. On the long-sequence dataset Mini-ImageNet, the BRL-Attention achieves the best performance among all groups; in this case, 24GB of memory is insufficient to run LDS and full-attention due to their heavy computation on learning/approximating global structures. Overall, the experiment suggests that the $k$-NN graphs are not necessarily informative, and besides, the BRL-Attention can learn useful latent graph structures from data while maintaining a memory-efficient nature.

## A.2    Complexity Evaluation on Real-World Textual Dataset

Table 6: Memory usage comparison on autoregressive training and inference under various context length.

| Model | Params | $n = 128$ | $n = 256$ | $n = 512$ | $n = 1024$ | $n = 2048$ | $n = 3072$ | $n = 4096$ |
|---|---|---|---|---|---|---|---|---|
| | | | | Autoregressive train | | | | |
| Full-Attention | 50.93M | 3.030GB | 3.301GB | 3.921GB | 5.760GB | 12.772GB | 23.869GB | OOM(>24GB) |
| Local-Attention | 50.93M | 3.030GB | 3.215GB | 3.563GB | 4.076GB | 5.740GB | 7.131GB | 8.732GB |
| BRL-Attention | 80.96M | 3.809GB | 3.979GB | 4.256GB | 4.842GB | 6.428GB | 7.948GB | 9.615GB |
| | | | | Inference (no grad) | | | | |
| Full-Attention | 50.93M | 2.260GB | 2.260GB | 2.264GB | 2.266GB | 2.590GB | 3.676GB | 4.875GB |
| Local-Attention | 50.93M | 2.260GB | 2.260GB | 2.264GB | 2.266GB | 2.340GB | 2.950GB | 3.063GB |
| BRL-Attention | 80.96M | 2.397GB | 2.401GB | 2.446GB | 2.456GB | 2.735GB | 3.280GB | 3.588GB |

We provide results of our BRL-Attention compared to Local- and Full-Attention on WikiText-103 dataset under various context length (block size) $n \in [128, 2096]$. Specifically, we evaluate the model with $w = 128$ since as shown Sec. 3.2, BRL-$w = 128$-$m = 128$ already outperforms the standard Transformer. The memory

(history) length is the same as input length $n$. For the backbone settings, we employ a model of 6 layers, 8 heads, and 512 embed dimensions. In Tab. 6, we show the VRAM usage of different models. In comparison, the BRL variant scale linearly with $n$ similar to Local-Attention, while requires significantly less VRAM than Full-Attention when $n > 1024$ on both train and inference phase. In the time efficiency evaluation Tab. 7, we observe the same trend where the efficiency of BRL outperforms Full-Attention on longer sequences.

Table 7: Time efficiency comparison on decoder inference under various context length.

| Model | Params | $n = 128$ | $n = 256$ | $n = 512$ | $n = 1024$ | $n = 2048$ | $n = 3072$ | $n = 4096$ |
|---|---|---|---|---|---|---|---|---|
| Full-Attention | 50.93M | 4.24ms | 4.26ms | 4.40ms | 10.28ms | 29.19ms | 61.95ms | 92.27ms |
| Local-Attention | 50.93M | 4.24ms | 4.26ms | 4.37ms | 5.82ms | 9.54ms | 13.53ms | 18.19ms |
| BRL-Attention | 80.96M | 10.26ms | 10.26ms | 10.98ms | 11.15ms | 14.10ms | 19.57ms | 24.85ms |

### A.3   On Larger Textual Dataset

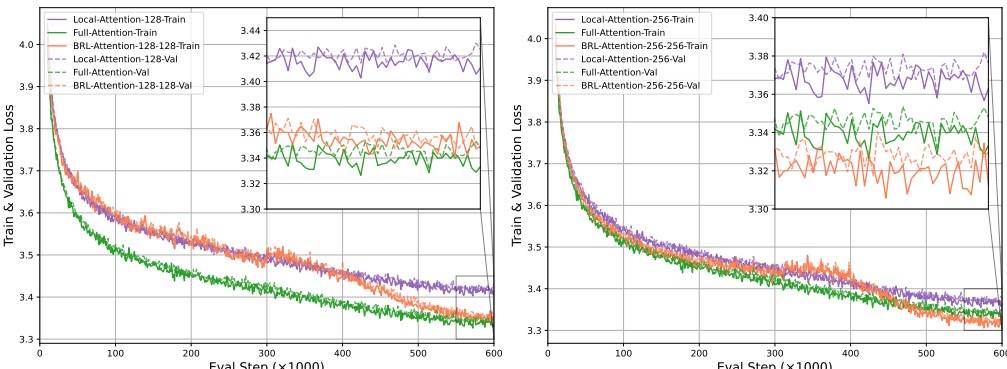

Figure 10: OpenWebText train and validation loss over steps. (**left**) The window size and compression length are both 128; (**right**) The window size and compression length are both 256.

In addition to WikiText-103, we study the autoregressive language pretraining on OpenWebText (Gokaslan et al., 2019; Radford et al., 2019), which is a scaled up datasets with ∼9B/4M tokens in train/validation set. We detail the model parameter configuration in Tab. 14.

In Fig. 10, we compare the train and validation losses of BRL-attention to local/full-attention. We employ the setup of and $m = w = 128$ and $m = w = 256$ ($\ll$ block size 1024), and employ the best practice as Sec. 3.2 where $\beta = 0.5$ and $\lambda_{\text{init}} = 0.5$. Similar to the results in Fig. 7, the BRL-attention with $m = w = 128$ can achieve comparable performance against full-attention, and BRL with $m = w = 256$ can achieve better results against full-attention while the losses of local-attention under both $w = 128$ and 256 deviate. This demonstrates that our method can scale with larger dataset and is effective on reducing the attention bottleneck.

## B   Theoretical Computational Complexity Analysis

### B.1   Inference Memory Complexity Analysis

We write $n$ for sequence length, $B$ for batch size, $H$ for the number of heads, and $L$ for the number of layers. We let $d = d_{\text{model}} = d_{\text{ffn}}$ for simplicity. Then the time complexity of vanilla Transformers $\mathcal{O}((3Bnd + BHn^2)L)$ according to (Kitaev et al., 2020), which is briefly $\mathcal{O}(n^2)$ considering the constant nature of $B, H, L, d$. Write $m$ for compressed token sequence length, assume $\phi(\mathbf{x}^{(l)}\mathbf{W}_q)\mathbf{W}_x = \mathbf{x}^{(l)}\mathbf{W}_x$ (by assuming the linearity of $\phi$) for simplicity, our proposed regularizer $\mathcal{F}_{\text{prop}}$ then consists of three major parts: (1) Eq. (9) requires respectively $Bnd + 2Bmd$ and $Bnm$ computation of feature transformations and of the

Table 8: Time/Memory efficiency comparison of various efficient transformers. N/A entries are for encoder-only models. $^\star$Efficient memory complexity might not equate a faster or more efficient model in practice.

| Methods | Memory Complexity | Time Complexity | Decode | Autoregressive Time Complexity | Score Matrix | Bottlenecked |
|---|---|---|---|---|---|---|
| Transformer | $\mathcal{O}(n^2)$ | $\mathcal{O}(n^2)$ | Yes | $\mathcal{O}(n^2)$ | Explicit | No |
| Sparse Transformer | $\mathcal{O}(n\sqrt{n})$ | $\mathcal{O}(n\sqrt{n})$ | Yes | $\mathcal{O}(n\sqrt{n})$ | Explicit | Potentially |
| Longformer (Local-Attention) | $\mathcal{O}(n)$ | $\mathcal{O}(n)$ | Yes (w/o Global Attn) | $\mathcal{O}(nd)$ | Explicit | Potentially |
| ETC | $\mathcal{O}(n)$ | $\mathcal{O}(n)$ | No | N/A | Explicit | No |
| BigBird | $\mathcal{O}(n)$ | $\mathcal{O}(n)$ | No | N/A | Explicit (Random) | No |
| Reformer | $\mathcal{O}(n\log n)$ | $\mathcal{O}(n\log n)$ | Yes | $\mathcal{O}(n\log n)$ | Explicit | Potentially |
| Synthesizer | $\mathcal{O}(n^2)$ | $\mathcal{O}(n)$ | Yes | $\mathcal{O}(n^2)$ | Explicit | Potentially |
| Performer | $\mathcal{O}(nc^2)$ | $\mathcal{O}(nc^2)$ | Yes | $\mathcal{O}(n^2c^2)$ (`cumsum`) | Implicit | N/A |
| Linformer | $\mathcal{O}(n)$ | $\mathcal{O}(n)$ | No | N/A | Implicit | N/A |
| BRL-Former (Ours) | $\mathcal{O}(n)$ | $\mathcal{O}(n)$ | Yes | $\mathcal{O}(n)$ | Partially Explicit | No |

$n \times m$ dimensional attention logits. (2) Eq. (7) requires $Bmd_{\text{ct}}$ for skip connection $\mathbf{M}$ and (3) $\mathcal{F}_{\text{comp}}$ requires $3Bnd_{\text{ct}}$ for token encoding, $Bmd_{\text{ct}}$ for compressed token encoding and $2Bmn$ for attention logits. In total, the complexity is $\mathcal{O}((Bnd+2Bmd+Bnm+Bmd_{\text{ct}}+3Bnd_{\text{ct}}+Bmd_{\text{ct}}+2Bmn)L) \approx \mathcal{O}(n)$ which shows regularizer $\mathcal{F}_{\text{prop}}$ scales linearly with $n$. Under Thm. 2.9, $\mathcal{F}_{\text{BRL}} = \mathcal{F}_{\text{gen}}(\cdot; D) + \mathcal{F}_{\text{prop}}$ approximates the expressibility of any continuous functions. Hence choose any sparse $D$, *e.g.* the attention with blockwise/strided patterns[3] (Beltagy et al., 2020), that gives a $\mathcal{O}(n)$ complexity $\mathcal{F}_{\text{gen}}(\cdot; D)$, then $\mathcal{F}_{\text{BRL}}$ is theoretically of approximately $\mathcal{O}(n)$ complexity.

### B.2 Training Memory Complexity Analysis

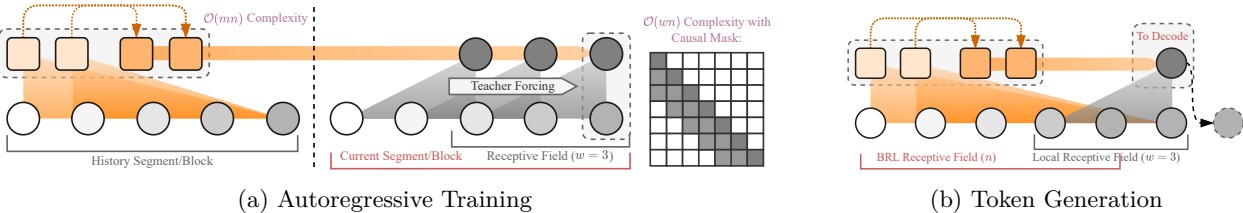

(a) Autoregressive Training  (b) Token Generation

Figure 11: To facilitate linear complexity autoregressive training, the compressed token for the current segment/block is derived from history tokens in former blocks such that it does not violate the causal structure of the current block. On generation, all in-context tokens are employed for computing compressed tokens.

The regularizer essentially approximates the kernalized self-attention under Prop. 2.4, 2.6. One may wonder what hindered us from explicitly formulating $\mathcal{F}_{\text{prop}}$ as kernalized self-attention since they are of the same inference complexity. To justify, we have already shown in Prop. 2.7 and Rmk. 2.8 that M2 applied to $\mathcal{F}_{\text{prop}}$ gives wider sensitivity bound while offering better robustness on noisy long input. Below, we give another viewpoint that focuses on the memory cost of *autoregressive training* compared to the prominent kernalized approach – Performer (Choromanski et al., 2020).

The complexity of Performer encoder-only training (bidirectional FAVOR) is of $\mathcal{O}(cd + nd + cn) \approx \mathcal{O}(n)$. While as noted in (Tay et al., 2022; Hua et al., 2022), the unidirectional variations **cannot** be causally masked in an efficient linear-time fashion. Training Performer for autoregressive tasks, which rely on parallelization and teacher forcing, requires a sequential left-to-right scan similar to RNNs. This makes it significantly slower under the hard requirement for manifesting the $c \times c$ KV matrix at every time step, recovering a $\mathcal{O}(n^2)$ complexity model.

For our BRL-Attention (with pattern $D$ explicitly set as (Dialated-)Sliding Window), the largest matrix constructed via $\mathcal{F}_{\text{gen}}(\cdot; D)$ is the $n \times w$ QK matrix where $w$ is the size of the window. Being compatible

---

[3]For sliding window attention (Beltagy et al., 2020), each token attends to $w$ tokens within the window size, hence per-layer computation is $\mathcal{O}((Bnd + BHnw)L)$. Supposedly, sliding window attention scales linearly with sequence length, which can be viewed as $\mathcal{O}(n)$ complexity.

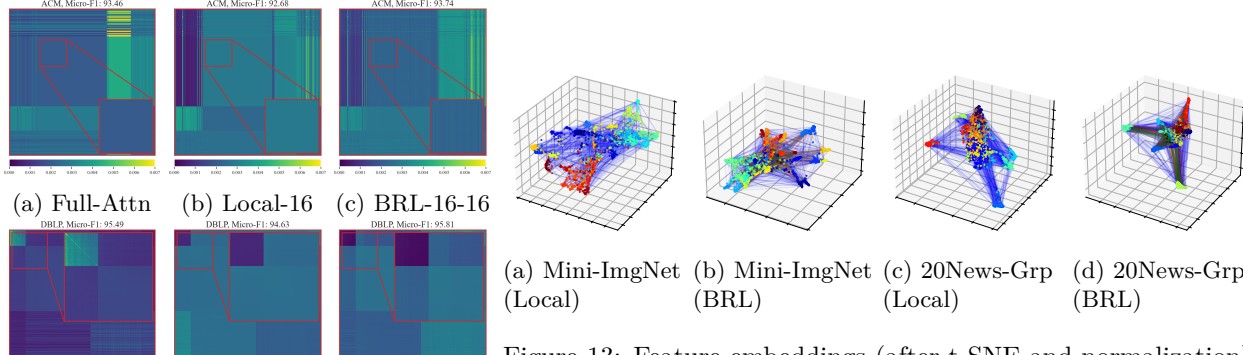

(a) Full-Attn  (b) Local-16  (c) BRL-16-16

(d) Full-Attn  (e) Local-16  (f) BRL-16-16

Figure 12: Visualizations of reconstructed attention maps. We sample a subset of sequences where the attentions from all heads are averaged.

(a) Mini-ImgNet (Local)  (b) Mini-ImgNet (BRL)  (c) 20News-Grp (Local)  (d) 20News-Grp (BRL)

Figure 13: Feature embeddings (after t-SNE and normalization) and edge connections produced by Local-Attention and BRL-Attention on graph-enhanced application datasets. We mark the nodes with a particular class with one color. The compressed tokens are in black color, red and green lines are, respectively, the attentive edges constructed by $\mathcal{F}_{\text{comp}}$ and $\mathcal{F}_{\text{prop}}$.

with causal masking, the memory complexity is reduced from $\mathcal{O}(n^2)$ to $\mathcal{O}(wn) \approx \mathcal{O}(n)$. For the BRL-Attention components, for $\mathcal{F}_{\text{prop}}$, we allow compressed tokens to attend to every token, for $\mathcal{F}_{\text{comp}}$, on teacher forcing $\mathbf{x}$, the input $\bar{\mathbf{x}}$ for $\mathbf{x}_{[\text{ct}]} \leftarrow \mathcal{F}_{\text{comp}}(\bar{\mathbf{x}})$ is the history block of $\mathbf{x}$ (where $\mathbf{x}$ is the current block), which does not violate the causal structure (see Fig. 11(a)). In this way, only a causal mask on the main token–to–main token interactions is necessary. Consequently, the largest matrix constructed in Eq. (9, 12) is $\max(\max(m, n) \times \max(d, d_{\text{ct}}), n \times m)$, which is also of $\approx \mathcal{O}(n)$ memory complexity considering $m$ and $d/d_{\text{ct}}$ are constant. Putting together, the BRL-Attention is of $\approx \mathcal{O}(n)$ complexity compared to the $\mathcal{O}(n^2)$ complexity of kernelized methods in training.

*Remark* B.1. As displayed earlier, the pattern-based attentions suffer from attention information bottlenecks when distant tokens cannot effectively communicate (Prop. 2.2). While adding *in-context global tokens* (Ainslie et al., 2020; Zaheer et al., 2020) potentially alleviates the bottleneck, it violates the causal structure, making them infeasible for autoregressive decoding (discussed in Sec. D.1). Kernel-based methods typically suffer from the quadratic complexity on autoregressive training (discussed in Sec. D.2) and can degrade performance if the kernel approximation is insufficient. In contrast, our method is not only capable of linear complexity autoregressive training (will be displayed in Sec. 3.1), but can approximate a model that is theoretically as expressive as full transformer (Thm. 2.9).

## C   Visualizations

**Visualization with Graphs.**   Fig. 12 displays averaged attention (reconstructed from $\mathbf{q}, \mathbf{k}$) matrices on ACM and DBLP, showing how each model variant connects different tokens. In contrast to local attention, which primarily focuses on neighborhood blocks and may overlook global structure, BRL-Attention consistently places varying weights on distant tokens. In Fig. 13, we visualize node embeddings (via t-SNE) and their attentively induced edges on 20News-Groups and Mini-ImageNet. The local-only variant assigns fewer inter-cluster edges, often concentrating on nodes within the same neighborhood. BRL-Attention instead increases cross-cluster edges, forming additional links that act as pivots for global propagation. By gathering and distributing context, the BRL-Attention with compressed tokens reduces over-squashing, allowing the model to learn better-separated node embeddings.

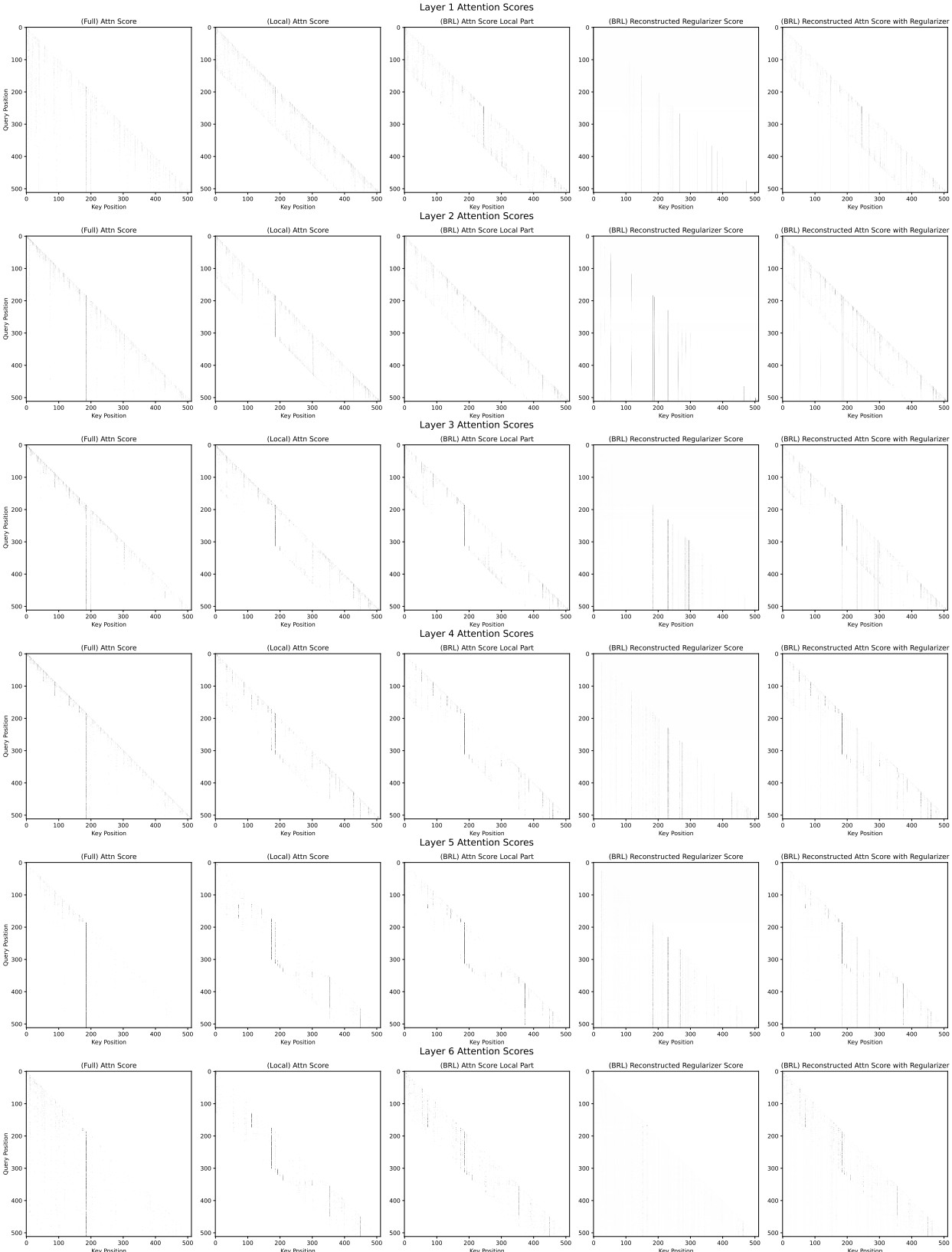

Figure 14: Attention score visualization (pretrained on WikiText-103). We compare Full, Local, and BRL-Attention. For Full and Local-Attention, respectively, we show the full attention score and the sliding window attention score. For BRL-Attention, we show the score from its local part, the reconstructed regularizer score, and their weighted (by coefficient $\lambda$) sum as reconstructed BRL-Attention score.

**Attention Score Visualization with Textual Dataset Pretrained Model.** In Fig. 14, we show the attention of different Transformer baselines pretrained on WikiText-103. In specific, we compare Full, Local, and BRL-Attention. For Full and Local-Attention, respectively, we show the full attn score and the sliding window attention score. For BRL-Attention, as the regularizer is affected directly to embedding rather than domain, it is difficult to visualize the actual additional information constructed by $\mathcal{F}_{\text{prop}}$. To give a indirect measurement of supplemental information flow, we show the score from its local part, the *Reconstructed Regularizer Score (RRS)*, and their weighted (by coefficient $\lambda$) sum as reconstructed BRL-Attention score. Particularly, the RRS is derived via the score in propagation map, averaging all compressed tokens:

$$(\mathbf{S}_{\text{RRS}})_{i,j} = (\mathbf{S}_{\text{prop}} \cdot \mathbf{S}_{\text{comp}})_{i,j}, \tag{15}$$

where $j \leq i$, and $\mathbf{S}_{\text{prop}}$ is the $n \times m$ score $\lambda \sigma_{\text{attn}}\left(\frac{\mathbf{z}^Q(\mathbf{z}^K)^\top}{\sqrt{d}}\right)$ in Eq. (9) and $\mathbf{S}_{\text{comp}}$ is the $m \times n$ score $(\mathbf{S}^{K1} - \gamma \mathbf{S}^{K2})$ in Eq. (12), denoting the propagation of information through compression mapping and propagation mapping on main tokens. In Fig. 14, we can observe that for some queries, the keys to attend (vertical lines) align with that of the full-attention. This indicate that the regularizer is learning useful information from global context, and can properly inject information to the receptive field-limited embeddings.

# D  Final Remarks

## D.1  On the Infeasibility of Global Token for Autoregressive Decoding

In autoregressive decoding, at time step $t$, a token $\mathbf{x}_t$ should only attend to the tokens from the past (*i.e.*, $\mathbf{x}_1, \mathbf{x}_2, \ldots, \mathbf{x}_{t-1}$), but *not* future tokens (*i.e.*, $\mathbf{x}_{t+1}, \mathbf{x}_{t+2}, \ldots, \mathbf{x}_n$). This is enforced through *causal masking* during the attention computation. Recall the attention operation in an autoregressive model defined as: $\mathbf{S}_{ij} = f_{\text{Softmax}}\left(\mathbf{q}_i \mathbf{k}_j^\top / \sqrt{d}\right)$. In autoregressive mode, the attention matrix $\mathbf{S}$ must be causal, meaning that $\forall i < j : \mathbf{S}_{ij} = 0$. This ensures that the model only attends to previous tokens and not future ones. The masked positions $\mathbf{S}_{ij}$ are set to zero for future tokens to avoid peeking.

Longformer, ETC and BigBird introduced **global tokens** to help scale attention for long sequences. These global tokens are selected tokens (in context) that can attend to every token in the sequence, regardless of the token position. For example, suppose we introduce a global token $\mathbf{x}_g$. In the attention matrix for a sequence $\{\mathbf{x}_1, \mathbf{x}_2, \ldots, \mathbf{x}_N, \mathbf{x}_g\}$, the global token $\mathbf{x}_g$ will attend to all tokens in the sequence: $\forall j \in [1, n] : \mathbf{S}_{g,j} = 1$.

However, in the autoregressive mode, we want each token $\mathbf{x}_t$ to attend only to previous tokens $\{\mathbf{x}_1, \ldots, \mathbf{x}_{t-1}\}$. However, with the global token in play, the model has to compute an attention matrix that includes cross-token interactions, the resulting attention matrix for the sequence $\{\mathbf{x}_1, \mathbf{x}_2, \ldots, \mathbf{x}_t, \mathbf{x}_g\}$ would look like: $\mathbf{S} = \begin{bmatrix} \mathbf{0}_{t \times t} & \mathbf{1}_{t \times 1} \\ \mathbf{1}_{1 \times t} & 1 \end{bmatrix}$, where the global token $\mathbf{x}_g$ interacts with all tokens, violating the causal structure.

## D.2  On the Quadratic Nature of (Linearized) Kernel Attention for Autoregressive Training

Recall the $c \times c$ KV matrix in Linearized Attention (Performer, Linear Transformer, Linformer etc.) is constructed by $\widehat{\mathbf{S}} = \mathbf{k}^\top \mathbf{v}$. Re-arranging the computation reduces the complexity w.r.t $n$ from quadratic to linear. In autoregressive decoding (generation), at time step $t$, define $\widehat{\mathbf{S}}_t = \mathbf{k}_{:t}^\top \mathbf{v}_{:t}$, notice that the computation of $\widehat{\mathbf{S}}_t$ can be fully incremental, *i.e.*, $\widehat{\mathbf{S}}_t = \widehat{\mathbf{S}}_{t-1} + \mathbf{k}_t^\top \mathbf{v}_t$. This means we only need to maintain a cache with constant $\mathcal{O}(c^2)$ memory and whenever a new input arrives at time stamp $t$, only constant $\mathcal{O}(c^2)$ computation is required to accumulate $\mathbf{k}_t^\top \mathbf{v}_t$ into $\widehat{\mathbf{S}}_{t_1}$ and get $\widehat{\mathbf{S}}_t$.

However, on autoregressive training (with teacher forcing), re-arranging the computation in linearized attention leads to a severe inefficiency. Due to the causal constraint for auto-regressive training, the query vector at each time step $\mathbf{q}_t$ corresponds to a different cache value $\widehat{\mathbf{S}}_t = \mathbf{k}_{:t}^\top \mathbf{v}_{:t}$. This requires the model to compute and cache $n$ different values $\{\widehat{\mathbf{S}}_t\}_{t=1}^n$ instead of only one $\mathbf{k}^\top \mathbf{v}$ in the non-autoregressive mode. In theory, the sequence $\{\widehat{\mathbf{S}}_t\}_{t=1}^n$ can be obtained in $\mathcal{O}(nc^2)$ by first computing $\{\mathbf{k}_t^\top \mathbf{v}_t\}_{t=1}^n$ and then performing a large cumulative sum (`cumsum`) over $n$ tokens. But in practice, the `cumsum` introduces an RNN-style sequential dependency of $n$ steps, where an $\mathcal{O}(c^2)$ state needs to be processed each step. The sequential dependency

not only limits the degree of parallelism, but more importantly requires $n$ memory access in the hard loop, which increase the complexity to quadratic.

### D.3 Differential Form in Compression Mapping Improves Attention SNR

To justify the impact of M2 on Signal-to-Noise Ratio (SNR) in attention mechanisms, we first categorize the keys into two sets relative to a given query: *relevant keys* and *irrelevant keys*. Let $\mathbf{k} = \{\mathbf{k}_1, \mathbf{k}_2, \ldots, \mathbf{k}_n\}$ be the set of all keys. Let $\mathbf{k}_{\text{rel}} \subseteq \mathbf{k}$ be the set of relevant keys, and $\mathbf{k}_{\text{irr}} = \mathbf{k} \setminus \mathbf{k}_{\text{rel}}$ be the set of irrelevant keys. Let $\mathbf{S} = \{\mathbf{s}_1, \mathbf{s}_2, \ldots, \mathbf{s}_n\}$ be the attention weights corresponding to keys in $\mathbf{k}$, where $\mathbf{s}_i$ is the attention weight for key $\mathbf{k}_i$.

**Definition D.1** (Attention SNR). We define the Attention SNR as the ratio of the average attention weight assigned to relevant keys to the average attention weight assigned to irrelevant keys:

$$r_{\text{attn}} = \frac{1}{|\mathbf{k}_{\text{rel}}|} \sum_{\mathbf{k}_j \in \mathbf{k}_{\text{rel}}} \mathbf{s}_j \Big/ \frac{1}{|\mathbf{k}_{\text{irr}}|} \sum_{\mathbf{k}_l \in \mathbf{k}_{\text{irr}}} \mathbf{s}_l, \tag{16}$$

where $|\mathbf{k}_{\text{rel}}|$ and $|\mathbf{k}_{\text{irr}}|$ are the number of relevant and irrelevant keys respectively. If $|\mathbf{k}_{\text{irr}}| \to 0$, we can consider SNR to be infinitely high, indicating perfect attention. If $|\mathbf{k}_{\text{rel}}| \to 0$ while $|\mathbf{k}_{\text{irr}}| > 0$, SNR is zero, indicating no signal. A higher SNR indicates a better ability of the attention mechanism to focus on relevant information while suppressing irrelevant information.

**Attention SNR on full and differential** $\mathcal{F}_{\text{comp}}(\mathbf{x}_{[\text{ct}]}; \mathbf{x})$. Ignoring the feature mappings in Eq. (12), the $\mathbf{S}^K$ is constructed by $\mathbf{q}_{[\text{ct}]} \in \mathbb{R}^{c \times d}$ and $\mathbf{k} \in \mathbb{R}^{n \times d}$ where the former is from compressed tokens and the latter is from main tokens. Denote the similarity scores in attention $\mathbf{S}^K$ as $\text{sim}(\mathbf{q}_{[\text{ct}]}, \mathbf{k})$, where $\mathbf{a}_i^K = \text{sim}(\mathbf{q}_{[\text{ct}]}, \mathbf{k}_i)$. Attention weight $\mathbf{S}^K$ is then formulated as $\mathbf{s}_i^K = \frac{\exp(\mathbf{a}_i^K)}{\sum_{j=1}^n \exp(\mathbf{a}_j^K)}$. Now, assume there is only one relevant key and one irrelevant key, let $\mathbf{a}_{\text{rel}}^K = a + \delta$ and $\mathbf{a}_{\text{irr}}^K = a$ (where $a$ is the base similarity and $\delta$ is a small perturbation). Then the SNR is expressed as

$$r_{\text{attn}}^{\text{full}} = \frac{\mathbf{s}_{\text{rel}}^K}{\mathbf{s}_{\text{irr}}^K} = \frac{\frac{\exp(a+\delta)}{\exp(a+\delta)+\exp(a)}}{\frac{\exp(a)}{\exp(a+\delta)+\exp(a)}} = \exp(\delta). \tag{17}$$

For the differential case, we have attention scores $\mathbf{S}^{K1}$ and $\mathbf{S}^{K2}$ respectively constructed by similarity scores $\mathbf{a}^{K1} = \text{sim}(\mathbf{q}_{[\text{ct}]}, \mathbf{k}^{K1})$ and $\mathbf{a}^{K2} = \text{sim}(\mathbf{q}_{[\text{ct}]}, \mathbf{k}^{K2})$. Similar to full-attention, let $\mathbf{a}_{\text{rel}}^{K1} = a + \delta$, $\mathbf{a}_{\text{irr}}^{K1} = a$ which exactly mimic the situation in full-attention, then let $\mathbf{a}_{\text{rel}}^{K2} = a + \delta_1$, $\mathbf{a}_{\text{irr}}^{K2} = a + \delta_2$ where $\delta_{\{1,2\}}$ are learned shift parameters by M2. Then with $\mathbf{s}_i^K = \frac{\exp(\mathbf{a}_i^K)}{\sum_{j=1}^n \exp(\mathbf{a}_j^K)}$, we have

$$r_{\text{attn}}^{\text{diff}} = \frac{\mathbf{s}_{\text{rel}}^{K1} - \gamma \mathbf{s}_{\text{rel}}^{K2}}{\mathbf{s}_{\text{irr}}^{K1} - \gamma \mathbf{s}_{\text{irr}}^{K2}} = \frac{\frac{\exp(\delta)}{\exp(\delta)+1} - \gamma \frac{\exp(\delta_1)}{\exp(\delta_1)+\exp(\delta_2)}}{\frac{1}{\exp(\delta)+1} - \gamma \frac{\exp(\delta_2)}{\exp(\delta_1)+\exp(\delta_2)}}. \tag{18}$$

**Proposition D.2.** *Under $\delta_1 - \delta_2 \leq \delta$, $r_{\text{attn}}^{\text{diff}} \geq \exp(\delta) = r_{\text{attn}}^{\text{full}}$ always holds, as long as the scaling factor $\gamma$ is positive.*

*Proof.* Proved in Appendix E.6. $\qquad\square$

Therefore, improving the attention SNR suffices by learning learning a differential form where $\delta_1 - \delta_2 < \delta$. Compared to the fixed SNR in full-attention, the differential form as Eq. (12) offers more flexibility in counteracting the attention noise. For the propagation mapping $\mathcal{F}_{\text{prop}}$, since the keys are constructed by compressed tokens $\mathbf{x}_{[\text{ct}]}$ which is assumed to be permutation invariant, it would be meaningless to index the noise, hence no requirement for noise suppression.

### D.4 Additional Discussions on Recent Works

#### D.4.1 Relation to Native Sparse Attention

Native Sparse Attention (NSA) is a recent efficient attention mechanism proposed in (Yuan et al., 2025). Specifically, NSA can be classified as learnable pattern-based attention that modifies the $\mathcal{F}_{\text{gen}}(\cdot; D)$. As $\mathcal{F}_{\text{gen}}$ is independent to $\mathcal{F}_{\text{BRL}}$, NSA is fully compatible to our method. We omit the comparison here since NSA requires compiling a fused-attention kernel and is only optimized for hopper GPUs (*e.g.*, H100).

#### D.4.2 Relation to State Space Models

Recent work on *deep State Space Models* SSMs, most prominently S4 (Gu et al., 2021) and its successors S5 (Smith et al., 2022), DSS (Gupta et al., 2022), and Mamba (Gu & Dao, 2023), demonstrates that learned state–space layers can model dependencies across tens of thousands of steps while retaining $\mathcal{O}(n)$ complexity. These models encode a sequence by evolving a latent state according to linear time-invariant dynamics (plus lightweight nonlinearities), and can be executed either as a fast FFT-based convolution (encoder style) or as a recurrent update (decoder style). Empirically, S4 established new state of the art on LRA and other long-sequence benchmarks and outperforming many Transformer variants.

**Architectural Contrast with BRL-Attention.** While both SSMs and BRL-Attention target the quadratic space/time bottleneck, their mechanisms are orthogonal. BRL-Attention keeps the content-based routing of self-attention but reduces cost by mediating global exchange through a fixed, trainable pool of compressed tokens, thus preserving the Transformer's dynamic contextual interactions and autoregressive training. In contrast, SSMs dispense with attention entirely: information is mixed through fixed kernel convolutions or state updates whose influence depends on position rather than token content. This design gives SSMs excellent memory compression and extrapolation to sequence lengths far beyond training, yet it lacks the explicit query–key reweighting that underpins in-context reasoning and fine-grained token-to-token alignment in attention models.

**On the Fairness of Direct Comparison.** SSMs summarise the entire past into a *fixed-width* latent state, so the model cannot *re-query* earlier tokens on demand; once a detail is not encoded it is effectively forgotten. This makes SSMs ideal for tasks whose signal is *distributed and uniform* along the sequence (*e.g.* long-range copy, audio, or algorithmic path-finding) but less suited to problems hinging on *selective, content-dependent retrieval*, such as long-context QA or few-shot in-context learning, where a token must attend to a rare, distant cue. BRL-Attention retains query–key similarity scores (via its compressed-token bottleneck) and full causal masking, enabling sparse but crucial interactions at generation time. Moreover, since BRL's goal is to **close the gap between full self-attention and pattern-based linear attention**, direct comparison to fundamentally different paradigms like SSMs can be misleading; we therefore reference SSM results as context but focus quantitative evaluation on *attention*-based baselines.

#### D.4.3 Relation to Hybrid Architectures

Our proposed method (as illustrated in Fig. 4) can be viewed as a mixture of two paradigms within a single layer, *i.e.* the generalized attention section and compressed token update section, which relates this paper to the field of hybrid models. Hybrid models generally aim to achieve a better trade-off between performance, computational efficiency, and long-context modeling than either approach can achieve alone.

**Transformer-Recurrent Hybrids.** This category uses Transformer attention as the primary building block and SSM/RNN parts as auxiliary. For instance, Griffin (De et al., 2024) mix repeating *recurrent blocks* (similar to GSS (Mehta et al., 2022)) with local/global Multi-Query Attention (MQA) blocks across different residual blocks, matching the performance of Full-Transformer despite being trained on over 6× fewer tokens. Similar models, *e.g.*, (Cao et al., 2024) combine LSTM layers with Transformer parts to get good sequence handling and context understanding; FLASH (Hua et al., 2022) split the context into chunks, letting full transformer to process each chunk and RNN to handle chunk wise relationships. However, bringing in sufficiently many Transformer layers will also introduce quadratic complexity. Moreover, while

RNNs are efficient for inference due to their sequential state updates, this very sequentiality makes training parallelization across time steps challenging.

**Transformer-SSM Hybrids.** These models are fundamentally built upon SSM or long convolution principles but may incorporate attention-like features or be used alongside attention in broader architectures. For instance, Jamba (Lieber et al., 2024) alternates between blocks of Transformer layers and Mamba layers. A Jamba block has $1:7$ ratio Attention-to-Mamba layers, which is proven to achieve optimal performance in the NLP domain (Waleffe et al., 2024). This setup lets Mamba manage long sequences of information, while the attention layers focus on specific token-level interactions. Simialrly, HybriDNA (Ma et al., 2025) also insert $1\times$ of Transformer block into $7\times$ of Mamba blocks for effectively balancing the advantages of both block types. Hyena (Poli et al., 2023), instead of using an SSM approach like Mamba, hybridizes implicitly parameterized long convolutions with data-controlled gating (mimicking the input-dependent nature of self-attention). In essence, these models are typically good for modeling extremely long sequences. However, fixed state memory can be a limitation in practical applications: once a token is processed, it's either included in the state memory or ignored, making it challenging to re-query an ignored token. This results in a lack of precision on tasks requiring strong associative recall, *e.g.*, copy tasks.

**Attention-Centric Hybrids.** The BRL-Attention modifies the attention operation itself to become globally aware and efficient, rather than replacing entire Transformer blocks with distinct SSM or RNN blocks. It avoids vertically stacking different layer types, which can be computationally heavy as each distinct full-sequence layer that processes the entire sequence embedding. Instead, BRL-Attention augments a base linear generalized attention mechanism with a global context mechanism (the compression-propagation). This addition incurs only marginal computational cost (as shown in Sec. A.2). To enable such guidance of global information, BRL-Attention must employs distinct strategies for training and generation in autoregressive scenarios. During training, the compressed tokens are updated using an attention-based method that learns to retrieve information from the available context, including a memory buffer of past tokens. During generation, this mechanism allows for step-by-step processing where the compressed tokens act as an evolving global summary, informed by the incrementally growing context, thus maintaining causal consistency.

### D.5 Discussion on Limitation

While BRL-Attention offers distinct advantages for processing long sequences, its relative benefits diminish for inputs of shorter length. For instance, Vision Transformers (ViTs) frequently operate on sequences composed of a limited number of patch-based tokens (*e.g.,* approximately 197 tokens for ViT-B/16). In such scenarios, the computational overhead of standard quadratic self-attention is often manageable. Consequently, the introduction of BRL-Attention's compressed tokens might yield only marginal performance gains while potentially adding system complexity. For sequences comprising only a few hundred tokens, conventional full self-attention or alternative, simpler efficient attention mechanisms may prove more practical.

Additionally, an empirical evaluation of speech processing tasks was not undertaken in the current work. State-of-the-art Transformer-based models in speech recognition and synthesis typically rely on extensive pre-training on very large-scale speech corpora. Such pre-training endeavors demand considerable computational resources, often spanning weeks or months on multi-GPU setups, which were beyond the scope and available resources for this study.

## E  Proofs and Derivations

### E.1  Proof of Prop. 2.2

*Proof.* Recall the update of token according to the sliding window attention

$$\mathbf{x}_i^{(l)} = \sum_{j=1}^{n} \mathbf{S}_{ij}^{\text{sw}} \mathbf{v}_j^{(l-1)} = \sum_{j_1=1}^{n} \mathbf{S}_{ij_1}^{\text{sw}} \mathbf{x}_{j_1}^{(l-1)} \mathbf{W}_v^{(l-1)}, \tag{19}$$

which can be expanded as

$$\mathbf{x}_i^{(l)} = \sum_{j_1=1}^n \mathbf{S}_{ij_1}^{\text{sw}}((\sum_{j_2=1}^n \mathbf{S}_{j_1 j_2}(\mathbf{x}_{j_2}^{(l-2)}\mathbf{W}_v^{(l-2)}))\mathbf{W}_v^{(l-1)}) \tag{20}$$

$$= \sum_{j_1=1}^n \sum_{j_2=1}^n (\mathbf{S}_{ij_1}^{\text{sw}}\mathbf{S}_{j_1 j_2})(\mathbf{x}_{j_2}^{(l-2)}\mathbf{W}_v^{(l-2)}\mathbf{W}_v^{(l-1)}) \tag{21}$$

Expanding until $\mathbf{x}_p^{(0)}$, we have $\mathbf{x}_i^{(l)}$ equal to

$$\sum_{j_1=1}^n \cdots \sum_{j_l=1}^n (\mathbf{S}_{ij_1}^{\text{sw}}\mathbf{S}_{j_1 j_2}^{\text{sw}}\cdots \mathbf{S}_{j_{l-1}j_l}^{\text{sw}})(\mathbf{x}_{j_l}^{(0)}\mathbf{W}_v^{(0)}\cdots \mathbf{W}_v^{(l-1)}). \tag{22}$$

Therefore, the derivative can be expressed as

$$\frac{\partial \mathbf{x}_i^{(L)}}{\partial \mathbf{x}_p^{(0)}} = \sum_{j_1\cdots j_{L-1}} \mathbf{S}_{ij_1}^{\text{sw}}\mathbf{S}_{j_1 j_2}^{\text{sw}}\cdots \mathbf{S}_{j_{L-1}p}^{\text{sw}}(\mathbf{W}_v^{(0)}\cdots \mathbf{W}_v^{(L-1)}) \tag{23}$$

$$= \sum_{\text{all paths from } p \text{ to } i} (\prod_{l=0}^{L-1} \mathbf{S}_{j_l j_{l+1}}^{\text{sw}}\mathbf{W}_v^{(l)}) \tag{24}$$

$$= ((\mathbf{S}^{\text{sw}})^L)_{ip}(\prod_{l=0}^{L-1} \mathbf{W}_v^{(l)}) \tag{25}$$

where the bound is

$$\|\frac{\partial \mathbf{x}_i^{(L)}}{\partial \mathbf{x}_p^{(0)}}\| = \|((\mathbf{S}^{\text{sw}})^L)_{ip}(\prod_{l=0}^{L-1} \mathbf{W}_v^{(l)})\| \tag{26}$$

$$= \|((\mathbf{S}^{\text{sw}})^L)_{ip}\|\|(\prod_{l=0}^{L-1} \mathbf{W}_v^{(l)})\| \tag{27}$$

$$\leq r_{\text{sw}}^L r_W^L \tag{28}$$

which suffice to derive how many layers are needed for token $p$ to reach $i$. For $\mathbf{S}^{\text{sw}}$, the maximum direct neighbor distance $d_{\text{mdnd}} = \frac{w+1}{2} - 1 = \frac{w-1}{2}$ where recall $w \geq 3$ is a odd number window size. Essentially, the $d_{\text{mdnd}}$ tells how far away we can "jump" up to from the current position in a single application of the adjacency. For token $i$ to commute with $j$ in the $L$-th layer, it must satisfy

$$Ld_{\text{mdnd}} \geq M \implies L \geq \frac{2M}{w-1}, \tag{29}$$

otherwise, $((\mathbf{S}^{\text{sw}})^L)_{ip}$ will be 0. Since $L$ is an integer, we take $L_{\min} = \lceil \frac{2M}{w-1} \rceil$ for obtaining the non-zero bound. Hence

$$\|\frac{\partial \mathbf{x}_i^{(L)}}{\partial \mathbf{x}_p^{(0)}}\| \leq \begin{cases} 0 & \text{if } L < \lceil \frac{2M}{w-1} \rceil \\ r_{\text{sw}}^L r_W^L & \text{if } L \geq \lceil \frac{2M}{w-1} \rceil \end{cases} \tag{30}$$

Replacing $L$ by $l+1$ concludes the proof. □

### E.2 Proof of Prop. 2.4

*Proof.* For softmax kernel, a choice of approximator $\phi$ is the $\phi_{\text{rfm}}$ defined as Eq. (10). With Eq. (3), under C1 and C2, the first input to $\mathcal{F}_{\text{prop}}$ is $\phi(\mathbf{q}) \approx \phi_{\text{rfm}}(\mathbf{q})$ and the second input is

$$\tilde{\mathbf{x}}_{[\text{ct}]}^{(l+1)} = [\sum_{j=1}^n \phi_{\text{rfm}}(\mathbf{k}_j^{(l)})^\top \|\text{flatten}(\sum_{j=1}^n \phi_{\text{rfm}}(\mathbf{k}_j^{(l)})^\top \mathbf{v}_j^{(l)})] \tag{31}$$

in $\mathbb{R}^{m \times d_{ct}} \equiv \mathbb{R}^{m \times c(1+d)}$. Then, instantiating the $\mathcal{F}_{\text{prop}}$ as

$$\mathcal{F}_{\text{prop}}(\mathbf{x}^{(l)}; \mathbf{x}_{[\text{ct}]}) = \frac{\phi(\mathbf{q}^{(l)})\text{expand}(\mathbf{x}_{[\text{ct}]}[:, c :])}{\phi(\mathbf{q}^{(l)})\mathbf{x}_{[\text{ct}]}[:, : c]} \tag{32}$$

$$= \frac{\phi_{\text{rfm}}(\mathbf{q}^{(l)})\text{expand}(\text{flatten}(\sum_{j=1}^{n} \phi_{\text{rfm}}(\mathbf{k}_j^{(l)})^{\top} \mathbf{v}_j^{(l)}))}{\phi_{\text{rfm}}(\mathbf{q}^{(l)}) \sum_{j=1}^{n} \phi_{\text{rfm}}(\mathbf{k}_j^{(l)})^{\top}} \tag{33}$$

$$= \frac{\phi_{\text{rfm}}(\mathbf{q}^{(l)}) \sum_{j=1}^{n} \phi_{\text{rfm}}(\mathbf{k}_j^{(l)})^{\top} \mathbf{v}_j^{(l)}}{\phi_{\text{rfm}}(\mathbf{q}^{(l)}) \sum_{j=1}^{n} \phi_{\text{rfm}}(\mathbf{k}_j^{(l)})^{\top}}, \tag{34}$$

gives the kernalized attention equivalent to $\mathcal{F}_{\text{kernel}}$. $\square$

### E.3 Proof of Prop. 2.6

*Proof.* By assumptions on input feature $\mathcal{X}$ and on transformation $\|\mathbf{W}\|$, we know that for all token of index $i$, $\mathbf{q}_i, \mathbf{k}_i, \mathbf{v}_i$ lies in a compact domain. As each component of $\phi$ is continuous, $\phi$ can be approximated arbitrarily well by MLP with $\mathcal{O}(1)$ width and depth (Cybenko, 1989). The continuity of $\phi$ also implies that $\phi(\mathbf{q}_i)$, $\sum_{j=1}^{n} \phi(\mathbf{k}_j)^{\top} \mathbf{v}_j$ lies in a compact domain, therefore the numerator lies in a compact domain. Lastly, since all operations do not involve $n$, the depth and width are constant in $n$. $\square$

### E.4 Proof of Prop. 2.7

*Proof.* We perform sensitivity analysis on the output of BR $\mathcal{F}_{\text{prop}}^{(l)}(\mathbf{x}^{(l)}, \mathbf{x}_{[\text{ct}]}^{(l+1)})_i$ of token $i$ after $(l)$-th layer propagation with respect to token $k$ by

$$\|\frac{\partial}{\partial \mathbf{x}_p^{(l)}} \mathcal{F}_{\text{prop}}^{(l)}(\mathbf{x}^{(l)}, \mathbf{x}_{[\text{ct}]}^{(l+1)})_i\| = \|\frac{\partial(\mathcal{F}_{\text{prop}}^{(l)})_i}{\partial \mathbf{x}_i^{(l)}} \frac{\partial \mathbf{x}_i^{(l)}}{\partial \mathbf{x}_p^{(l)}} + \frac{\partial(\mathcal{F}_{\text{prop}}^{(l)})_i}{\partial \mathbf{x}_{[\text{ct}]}^{(l+1)}} \frac{\partial \mathbf{x}_{[\text{ct}]}^{(l+1)}}{\partial \mathbf{x}_p^{(l)}}\| \tag{35}$$

$$= 0 + \|\underbrace{\frac{\partial(\mathcal{F}_{\text{prop}}^{(l)})_i}{\partial \mathbf{x}_{[\text{ct}]}^{(l+1)}}}_{\text{Term T1}}\| \|\underbrace{\frac{\partial \mathcal{F}_{\text{comp}}^{(l)}(\mathbf{x}^{(l)}, \mathbf{x}_{\text{ct}}^{(l)})}{\partial \mathbf{x}_p^{(l)}}}_{\text{Term T2}}\| \tag{36}$$

For term T1, let $\phi(\mathbf{q}) \equiv \mathbf{x}\mathbf{W}_\phi$, we simplify the formulation of $\mathcal{F}_{\text{prop}}$ in Eq. (9) as

$$[\mathcal{F}_{\text{prop}}^{(l)}]_i = \text{sm}(\frac{1}{\sqrt{d}} \mathbf{x}_i^{(l)} \mathbf{W}_\phi (\mathbf{x}_{[\text{ct}]}^{(l+1)} \mathbf{W}_z)^{\top}) \mathbf{x}_{[\text{ct}]}^{(l+1)} \mathbf{W}_z, \tag{37}$$

where $\text{sm}(\cdot)$ denote the row-wise Softmax function. Therefore, term T1 can be simplified as

$$\|\frac{\partial[\mathcal{F}_{\text{prop}}^{(l)}]_i}{\partial \mathbf{x}_{[\text{ct}]}^{(l+1)}}\| = \|\frac{\partial \text{sm}(\frac{1}{\sqrt{d}} \mathbf{x}_i^{(l)} \mathbf{W}_\phi (\mathbf{x}_{[\text{ct}]}^{(l+1)} \mathbf{W}_z)^{\top}) \mathbf{x}_{[\text{ct}]}^{(l+1)} \mathbf{W}_z}{\partial \mathbf{x}_{[\text{ct}]}^{(l+1)}}\| \tag{38}$$

$$= \|\frac{\partial \text{sm}(\frac{1}{\sqrt{d}} \mathbf{x}_i^{(l)} \mathbf{W}_\phi (\mathbf{x}_{[\text{ct}]}^{(l+1)} \mathbf{W}_z)^{\top})}{\partial \mathbf{x}_{[\text{ct}]}^{(l+1)}} \mathbf{W}_z\| \tag{39}$$

Let $H_q = \frac{1}{\sqrt{d}} \mathbf{x}_i^{(l)} \mathbf{W}_\phi \in \mathbb{R}^{1 \times d_{ct}}$ we have

$$\|\frac{\partial(\mathcal{F}_{\text{prop}}^{(l)})_i}{\partial \mathbf{x}_{[\text{ct}]}^{(l+1)}}\| = \|\frac{\partial \text{sm}(H_q (\mathbf{x}_{[\text{ct}]}^{(l+1)} \mathbf{W}_z)^{\top})}{\partial H_q (\mathbf{x}_{[\text{ct}]}^{(l+1)} \mathbf{W}_z)^{\top}} \frac{\partial H_q (\mathbf{x}_{[\text{ct}]}^{(l+1)} \mathbf{W}_z)^{\top}}{\mathbf{x}_{[\text{ct}]}^{(l+1)}} \mathbf{W}_z\| \tag{40}$$

$$= \|\mathcal{O}(1) \cdot \frac{1}{\sqrt{d}} \mathbf{x}_i^{(l)} \mathbf{W}_\phi \mathbf{W}_z^{\top} \mathbf{W}_z\| \tag{41}$$

$$\leq \mathcal{O}(\frac{r_x r_W^3}{\sqrt{d}}). \tag{42}$$

For term T2, let us first consider $\mathcal{F}_{\text{comp}}$ to be in simple cross-attention form (with scaling factor changed from $\sqrt{d}$ to $m\sqrt{d_{\text{ct}}}$)

$$\mathcal{F}_{\text{comp}}^{(l)} = \text{sm}(\frac{1}{m\sqrt{d_{\text{ct}}}}\mathbf{x}_{[\text{ct}]}^{(l+1)}\mathbf{W}_h^Q(\mathbf{x}^{(l)}\mathbf{W}_h^K)^\top)\mathbf{x}^{(l)}\mathbf{W}_h^V, \tag{43}$$

then similar to the derivations above, let $S_{\text{ct}}^{(l)} = \frac{1}{m\sqrt{d_{\text{ct}}}}\mathbf{x}_{[\text{ct}]}^{(l+1)}\mathbf{W}_h^Q(\mathbf{x}^{(l)}\mathbf{W}_h^K)^\top \in \mathbb{R}^{m \times n}$, we have

$$\|\frac{\partial \mathcal{F}_{\text{comp}}^{(l)}(\mathbf{x}^{(l)}, \mathbf{x}_{[\text{ct}]}^{(l)})}{\partial \mathbf{x}_p^{(l)}}\| = \|\frac{\partial \text{sm}(S_{\text{ct}}^{(l)})}{\partial S_{\text{ct}}^{(l)}}\frac{\partial S_{\text{ct}}^{(l)}}{\partial \mathbf{x}_p^{(l)}}\mathbf{W}_h^V\| \tag{44}$$

$$= \|\mathcal{O}(m) \cdot \frac{1}{m\sqrt{d_{\text{ct}}}}\mathbf{x}_{[\text{ct}]}^{(l+1)}\mathbf{W}_h^Q(\mathbf{W}_h^K)^\top\mathbf{W}_h^V\| \tag{45}$$

$$\leq \mathcal{O}(\frac{m}{m}\frac{r_{\text{ct}}r_W^3}{\sqrt{d_{\text{ct}}}}) = \mathcal{O}(\frac{r_{\text{ct}}r_W^3}{\sqrt{d_{\text{ct}}}}). \tag{46}$$

Hence with Eq. (46), the bound Eq. (36) is eventually

$$\|\frac{\partial}{\partial \mathbf{x}_p^{(l)}}\mathcal{F}_{\text{prop}}^{(l)}(\mathbf{x}^{(l)}, \mathbf{x}_{[\text{ct}]}^{(l+1)})_i\| \leq \mathcal{O}(\frac{r_x r_{\text{ct}}r_W^6}{\sqrt{dd_{\text{ct}}}}), \tag{47}$$

Now, if we consider $\mathcal{F}_{\text{comp}}$ to be in differential cross-attention form, briefly defined as

$$\mathcal{F}_{\text{comp}}^{(l)} = (\text{sm}(S_{K1}^{(l)}) - \gamma\text{sm}(S_{K2}^{(l)}))\mathbf{x}^{(l)}\mathbf{W}_h^V, \tag{48}$$

where $\mathbf{S}_{K1}^{(l)} = \frac{\mathbf{x}_{[\text{ct}]}^{(l+1)}\mathbf{W}_h(\mathbf{x}^{(l)}\mathbf{W}_h^{K1})^\top}{m\sqrt{d_{\text{ct}}}}$ and similarly for $\mathbf{S}^{K2}$. Then the bound can be derived as

$$\|\frac{\partial \mathcal{F}_{\text{comp}}^{(l)}(\mathbf{x}^{(l)}, \mathbf{x}_{[\text{ct}]}^{(l)})}{\partial \mathbf{x}_p^{(l)}}\| = \|(\frac{\partial \text{sm}(S_{K1}^{(l)})}{\partial S_{K1}^{(l)}}\frac{\partial S_{K1}^{(l)}}{\partial \mathbf{x}_p^{(l)}} - \gamma\frac{\partial \text{sm}(S_{K2}^{(l)})}{\partial S_{K2}^{(l)}}\frac{\partial S_{K2}^{(l)}}{\partial \mathbf{x}_p^{(l)}})\mathbf{W}_h^V\| \tag{49}$$

$$\leq \|\frac{\partial \text{sm}(S_{K1}^{(l)})}{\partial S_{K1}^{(l)}}\frac{\partial S_{K1}^{(l)}}{\partial \mathbf{x}_p^{(l)}}\mathbf{W}_h^V\| + |\gamma|\|\frac{\partial \text{sm}(S_{K2}^{(l)})}{\partial S_{K2}^{(l)}}\frac{\partial S_{K2}^{(l)}}{\partial \mathbf{x}_p^{(l)}}\mathbf{W}_h^V\| \tag{50}$$

$$\leq \mathcal{O}(\frac{r_{\text{ct}}r_W^3}{\sqrt{d_{\text{ct}}}}(1 + |\gamma|)), \tag{51}$$

plugging in Eq. (36) gives

$$\|\frac{\partial}{\partial \mathbf{x}_p^{(l)}}\mathcal{F}_{\text{prop}}^{(l)}(\mathbf{x}^{(l)}, \mathbf{x}_{[\text{ct}]}^{(l+1)})_i\| \leq \mathcal{O}(\frac{r_x r_{\text{ct}}r_W^6}{\sqrt{dd_{\text{ct}}}}(1 + |\gamma|)). \tag{52}$$

This concludes the proof. □

### E.5 Proof of Thm. 2.9

*Proof.* We firstly define the star-graph:

**Definition E.1.** The star-graph $S$ centered at 0 is the graph defined on $\{0, \ldots, n\}$. The neighborhood of all vertices $i$ is $\mathcal{N}(i) = \{0, i\}$ for $i \in \{1, \ldots, n\}$ and $\mathcal{N}(0) = \{1, \ldots, n\}$.

Define input $\mathbf{x} = \{\mathbf{x}_0, \mathbf{x}_1, \ldots, \mathbf{x}_n\} \in \mathbb{R}^{(n+1) \times d}$ where $\{\mathbf{x}\}_{i=1}^n$ are the main tokens and $\mathbf{x}_0$ is the center token introduced by star-graph $S$. Now, we define a simplified version of BRL-Attention using softmax attention and without differential form, where the compression mapping

$$\tilde{\mathbf{x}} = \mathcal{F}_{\text{comp}}(\mathbf{x}) = \frac{\sum_{j=1}^n \exp(\mathbf{q}_{[\text{ct}]}\mathbf{k}_j^\top)\mathbf{v}_j}{\sum_{k=1}^n \exp(\mathbf{q}_{[\text{ct}]}\mathbf{k}_k^\top)}, \tag{53}$$

where $\mathbf{q} = f_Q(\mathbf{x}), \mathbf{k} = f_K(\mathbf{x}), \mathbf{v} = f_V(\mathbf{x})$. Under column-wise softmax, a trivial propagation mapping can be expressed as

$$\mathcal{F}_{\text{prop}}(\mathbf{x}, \widetilde{\mathbf{x}})_i = \frac{\exp(\mathbf{q}_i \widetilde{\mathbf{k}}^\top)}{\sum_{k \in \mathcal{N}_D(i)} \exp(\mathbf{q}_k \widetilde{\mathbf{k}}^\top)} \widetilde{\mathbf{v}}, \tag{54}$$

where $i \in \{1, \ldots, n\}$. Notably, both $\mathcal{F}_{\text{comp}}$ and $\mathcal{F}_{\text{prop}}$ does not require the usage of $\mathbf{x}_0$. Then for

$$\mathcal{F}_{\text{BRL}} = \mathcal{F}_{\text{gen}}(\mathbf{x}_{1\ldots n}; D) + \mathcal{F}_{\text{prop}}(\mathbf{x}, \widetilde{\mathbf{x}}) \in \mathbb{R}^{n \times d}, \tag{55}$$

we have the following proposition.

**Proposition E.2.** *For any pattern $D$ such that $D \cap S = \emptyset$, the $\mathcal{F}_{\text{gen}}(\mathbf{x}_{0\ldots n}; D \cup S)_{1\ldots n}$ can be simulated by $\mathcal{F}_{\text{BRL}}$ arbitrarily well, where $D \cup S$ can be regarded as any graph containing star-graph $S$.*

*Proof.* Recall the generalized attention which computes

$$\mathcal{F}_{\text{gen}}^{(l)}(\mathbf{x}; D \cup S)_i = \sum_{j \in \mathcal{N}_{D \cup S}(i)} \frac{\exp(\mathbf{q}_i \mathbf{k}_j^\top)}{\sum_{k \in \mathcal{N}_{D \cup S}(i)} \exp(\mathbf{q}_i \mathbf{k}_k^\top)} \mathbf{v}_j, \tag{56}$$

for pattern $D \cup S$. Recall the definition of star-graph, for any token $i \in \{1, \ldots, n\}$ the neighborhood under $D \cup S$ is $\mathcal{N}(i) = \mathcal{N}_D(i) \cup \{0\}$. We now write the generalized attention output for index $i \in \{1, \ldots, n\}$ under the pattern $D \cup S$ as

$$\mathcal{F}_{\text{gen}}(\mathbf{x}; D \cup S)_i = \underbrace{\frac{\exp(\bar{\mathbf{q}}_i \bar{\mathbf{k}}_0^\top)}{Z_i} \bar{\mathbf{v}}_0}_{\text{Term T1}} + \underbrace{\sum_{j \in \mathcal{N}_D(i)} \frac{\exp(\bar{\mathbf{q}}_i \bar{\mathbf{k}}_j^\top)}{Z_i} \bar{\mathbf{v}}_j}_{\text{Term T2}}, \tag{57}$$

$$\text{where} \quad Z_i = \exp(\bar{\mathbf{q}}_i \bar{\mathbf{k}}_0^\top) + \sum_{k \in \mathcal{N}_D(i)} \exp(\bar{\mathbf{q}}_i \bar{\mathbf{k}}_k^\top). \tag{58}$$

We use $\bar{\mathbf{q}}, \bar{\mathbf{k}}, \bar{\mathbf{v}}$ instead of $\mathbf{q}, \mathbf{k}, \mathbf{v}$ to highlight that they are generated by different neural networks. Notice that T2 is exactly the contribution from pattern $D$, which is equivalantly $\mathcal{F}_{\text{gen}}(\mathbf{x}; D) = \mathcal{F}_{\text{gen}}(\mathbf{x}_{1\ldots n}; D)$ as $D \cap S = \emptyset$. Therefore, it suffice to show that T1 can be simulated by $\mathcal{F}_{\text{prop}}(\mathbf{x}, \widetilde{\mathbf{x}})_i$ for all $i \in \{1, \ldots, n\}$.

Observe the difference between T1 and Eq. (54) that the numerator $\exp(\bar{\mathbf{q}}_i \bar{\mathbf{k}}_0^\top)$ in T1 can be easily simulated by $\exp(\mathbf{q}_i \widetilde{\mathbf{k}}^\top)$ as $\widetilde{\mathbf{k}}$ aggregates the information from $\mathbf{x}_{\text{[ct]}}$ (which can be regarded as $\mathbf{x}_0$) by compression Eq. (53), and one may simply let $\mathbf{q}_i = \bar{\mathbf{q}}_i$. On the denominator, as $\exp(\bar{\mathbf{q}}_i \bar{\mathbf{k}}_0^\top)$ can be simulated, it suffice to show that $\sum_{k \in \mathcal{N}_D(i)} \exp(\bar{\mathbf{q}}_i \bar{\mathbf{k}}_k^\top)$ can be simulated by $\sum_{k \in \mathcal{N}_D(i)} \exp(\mathbf{q}_k \widetilde{\mathbf{k}}^\top)$. According to the property of inner product, $\mathbf{q}\mathbf{k}^\top$ produces a scalar, thus $\mathbf{q}\mathbf{k}^\top = \mathbf{k}\mathbf{q}^\top$. As $\forall k \in \mathcal{N}_D(i) : \bar{\mathbf{k}}_k$ can be exactly simulated by let $\forall k \in \mathcal{N}_D(i) : \mathbf{q}_k = \bar{\mathbf{k}}_k$, the problem reduced to proving $\widetilde{\mathbf{k}}$ can simulate any $\bar{\mathbf{q}}_i$. Notice that by the design of compression Eq. (53), $\widetilde{\mathbf{x}}$ aggregates information from all tokens $\{\mathbf{x}_i\}_{i=1}^n$, thus we may design $\widetilde{\mathbf{k}} = \widetilde{f_K}(\mathbf{x}, i)$ as a simple decoder, which receives the $i$-th token as input and decode the corresponding token $\mathbf{x}_i$ out of $\widetilde{\mathbf{x}}$. Concluding above, under $\mathbf{x}_{\text{[ct]}} = \mathbf{x}_0$ in Eq. (53) (which eliminate the use of $\mathbf{x}_0$ for star-graph), we have shown that $\mathcal{F}_{\text{gen}}^{(l)}(\mathbf{x}; D \cup S)$ can be simulated by $\mathcal{F}_{\text{BRL}} = \mathcal{F}_{\text{gen}}(\cdot; D) + \mathcal{F}_{\text{prop}}(\cdot; \mathcal{F}_{\text{comp}}(\cdot))$. $\qquad\square$

Note that, when only main tokens $\{\mathbf{x}_i\}_{i=1}^n$ are employed (just as in the BRL-Attention or any regular attentions), we naturally have $D \cap S = \emptyset$. Therefore, Prop. E.2 works for BRL-Attention with any pattern $D$. Next, we leverage the result in (Zaheer et al., 2020) to complete our proof.

**Theorem E.3** (Thm. 1 (Zaheer et al., 2020))**.** *Given $1 \leq p < \infty$ and $\epsilon > 0$, for any continuous functions $\mathcal{F}_{\text{con}} : [0, 1]^{n \times d} \to \mathbb{R}^{n \times d}$, there exists a transformer with sparse-attention, $\mathcal{F}_{\text{gen}}(\cdot; D \cup S)$ such that $d_p(\mathcal{F}_{\text{con}}, \mathcal{F}_{\text{gen}}(\cdot; D \cup S)) \leq \epsilon$ where $D \cup S$ is any graph containing star graph $S$.*

Combining the result from Prop. E.2 and Thm. E.3, we arrives at our conclusion that $d_p(\mathcal{F}_{\text{con}}, \mathcal{F}_{\text{gen}}(\cdot; D) + \mathcal{F}_{\text{prop}}) \leq \epsilon$ with *arbitrary* sparse-attention $D$. $\qquad\square$

### E.6 Proof of Prop. D.2

*Proof.* Given

$$r_{\text{attn}}^{\text{full}} = \frac{\mathbf{s}_{\text{rel}}^K}{\mathbf{s}_{\text{irr}}^K} = \frac{\frac{\exp(a+\delta)}{\exp(a+\delta)+\exp(a)}}{\frac{\exp(a)}{\exp(a+\delta)+\exp(a)}} = \exp(\delta), \tag{59}$$

and

$$r_{\text{attn}}^{\text{diff}} = \frac{\mathbf{s}_{\text{rel}}^{K1} - \gamma \mathbf{s}_{\text{rel}}^{K2}}{\mathbf{s}_{\text{irr}}^{K1} - \gamma \mathbf{s}_{\text{irr}}^{K2}} = \frac{\frac{\exp(\delta)}{\exp(\delta)+1} - \gamma \frac{\exp(\delta_1)}{\exp(\delta_1)+\exp(\delta_2)}}{\frac{1}{\exp(\delta)+1} - \gamma \frac{\exp(\delta_2)}{\exp(\delta_1)+\exp(\delta_2)}}, \tag{60}$$

let $A = \exp(\delta) + 1$ and $B = \exp(\delta_1) + \exp(\delta_2)$, then the SNR for differential form can be simplified as

$$r_{\text{attn}}^{\text{diff}} = \frac{\frac{\exp(\delta)}{A} - \gamma \frac{\exp(\delta_1)}{B}}{\frac{1}{A} - \gamma \frac{\exp(\delta_2)}{B}} = \frac{B \exp(\delta) - \gamma A \exp(\delta_1)}{B - \gamma A \exp(\delta_2)}. \tag{61}$$

When $r_{\text{attn}}^{\text{diff}} \geq r_{\text{attn}}^{\text{full}}$, assume $\gamma > 0$, we essentially have

$$\frac{B \exp(\delta) - \gamma A \exp(\delta_1)}{B - \gamma A \exp(\delta_2)} \geq \exp(\delta) \tag{62}$$

$$\iff -\gamma A \exp(\delta_1) \geq -\gamma A \exp(\delta) \exp(\delta_2) \tag{63}$$

$$\iff \exp(\delta_1) \leq \exp(\delta + \delta_2) \tag{64}$$

$$\iff \delta_1 \leq \delta + \delta_2, \tag{65}$$

which concludes the proof. □

### E.7 On the Lower-bound of Jacobian Bottleneck

We show in this section that under certain assumptions on singular value of weight matrix, the lower-bound of the Jacobian bottleneck in Prop. 2.7 is positive. Below, we give a proof on simplified model architecture.

*Proof.* We aim to demonstrate that the Jacobian corresponding to the regularization path in BRL-Attention can have a non-zero lower bound in general cases, thereby alleviating the information bottleneck observed in purely pattern-based attention mechanisms. Recall the Jacobian bottleneck established in Prop. 2.2 for pattern-based attention $\mathcal{F}_{\text{gen}}$, the proposition implies that when $l < \lceil \frac{2M}{w-1} \rceil - 1$, the Jacobian through $\mathcal{F}_{\text{gen}}$ is zero. Under these conditions, the Jacobian of the full BRL-Attention is $J_{\text{BRL}} = \frac{\partial \mathcal{F}_{\text{BRL}}^{(l)}}{\partial \mathbf{x}_p^{(l)}} = \frac{\partial \mathcal{F}_{\text{prop}}^{(l)}}{\partial \mathbf{x}_p^{(l)}}$, $(J_{\text{BRL}})_{ip} = \frac{\partial (\mathcal{F}_{\text{prop}}^{(l)})_i}{\partial \mathbf{x}_p^{(l)}}$. Next, we make the following assumptions to facilitate the justification that $J_{\text{BRL}} > 0$: *(1)* We assume $\beta = 0$ and $\lambda = 1$ for simplicity. *(2)* Non-singular weight matrices: That is, learnable weight matrices $\mathbf{W}$ (e.g., $\mathbf{W}_h^Q, \mathbf{W}_h^K, \mathbf{W}_h^V$ in $\mathcal{F}_{\text{comp}}$; $\mathbf{W}_{\mathbf{z}}^Q, \mathbf{W}_{\mathbf{z}}^K, \mathbf{W}_{\mathbf{z}}^V$ in $\mathcal{F}_{\text{prop}}$) along the computational path have their smallest singular value $\sigma_{\min}(\mathbf{W}) \geq s_W > 0$. *(3)* We assume for all $l > 0$, $\mathbf{x}_*^{(l)}$ and $(\mathbf{x}_{[\text{ct}]}^{(l)})_*$ are unit vectors.

Consider the Jacobian for $l$-th layer: $(J_{\text{BRL}})_{ip} = \frac{\partial (\mathcal{F}_{\text{prop}}^{(l)})_i}{\partial \mathbf{x}_p^{(l)}}$. By Proof. E.4 we have

$$(J_1)_{ik} = \frac{\partial (\mathcal{F}_{\text{prop}}^{(l)})_i}{\partial (\mathbf{x}_{[\text{ct}]}^{(l+1)})_k} = \frac{\partial \text{sm}(H_q((\mathbf{x}_{[\text{ct}]}^{(l+1)})_k \mathbf{W}_z)^\top)}{\partial H_q((\mathbf{x}_{[\text{ct}]}^{(l+1)})_k \mathbf{W}_z)^\top} \frac{\partial H_q((\mathbf{x}_{[\text{ct}]}^{(l+1)})_k \mathbf{W}_z)^\top}{(\mathbf{x}_{[\text{ct}]}^{(l+1)})_k} \mathbf{W}_z = \mathcal{O}(1) \cdot \frac{1}{\sqrt{d}} \mathbf{x}_i^{(l)} \mathbf{W}_\phi \mathbf{W}_z^\top \mathbf{W}_z. \tag{66}$$

By aforementioned assumptions, let $\mathbf{W}_\phi = \mathbf{W}_z = \mathbf{W}_z^{Q,K,V}$ for simplicity. Recall that for any matrix $\mathbf{M}$, its smallest singular value $\sigma_{\min}(\mathbf{M})$ is defined as: $\sigma_{\min}(\mathbf{M}) = \min_{\|\mathbf{u}=1\|}\|\mathbf{u}\mathbf{M}\|$. As $\mathbf{x}_i$ is assumed to be a unit vector, we have

$$\|(J_1)_{ik}\| = \|\mathbf{x}_i \cdot \mathcal{O}(1)\frac{1}{\sqrt{d}}\mathbf{W}_z\mathbf{W}_z^\top\mathbf{W}_z\| \geq \frac{1}{\sqrt{d}}\|\mathbf{x}_i\|\sigma_{\min}(\mathbf{W}_z)^3 \geq \frac{s_W^3}{\sqrt{d}}\|\mathbf{x}_i\|. \tag{67}$$

Therefore, we can say $\|J_1\| \geq \frac{s_W^3}{\sqrt{d}} > 0$. Next, we define $(J_2)_{kp} = \frac{\partial(\mathbf{x}_{[\texttt{ct}]}^{(l+1)})_k}{\partial\mathbf{x}_p^{(l)}} = \frac{\mathcal{F}_{\text{comp}}^{(l)}(\mathbf{x}_{[\texttt{ct}]}^{(l)}, \mathbf{x}^{(l)})_k}{\mathbf{x}_p^{(l)}}$, then with similar procedure, we can derive that $\|J_2\| \geq \frac{s_W^3}{\sqrt{d}} > 0$ since $(\mathbf{x}_{[\texttt{ct}]})_k$ is also assumed to be unit vectors. Therefore, according to the chain rule, we have

$$\|(J_{\text{BRL}})_{ip}\| = \|\sum_{k=1}^m (J_1)_{ik}(J_2)_{kp}\| \geq \frac{ms_W^6}{d} > 0. \tag{68}$$

This concludes the proof. $\qquad\square$

Although given the above proof, we still note that establishing a universally **strictly greater than zero** lower bound for sensitivity (information propagation) is generally problematic without any assumption, even for standard full-attention mechanisms. The actual flow of information, and thus the Jacobian, is inherently data-dependent and context-specific:

- **Data Specificity**: If, for a particular input sequence, there is no meaningful semantic relationship to be captured between two specific tokens $i$ and $p$, the attention scores connecting them (even in a full-attention model) might naturally be zero or near-zero. In such an instance, the Jacobian representing the influence of token $p$ on token $i$ would also be zero, leading to a zero lower bound.

- **On Full-Attention w/o Bottleneck**: Even in a Full-Attention Transformer, if the attention mechanism learns that token $i$ should not attend to token $p$ for a given context, the corresponding attention weight will be zero, and thus the propagated information (and its gradient) through that specific path will be zero.

Therefore, a lower bound on sensitivity would likely still be nearly zero in many valid scenarios. The crucial aspect of our BRL-Attention is not to guarantee a *minimum amount* of information flow between all token pairs at all times, but rather to remove the *structural certainty of zero flow* imposed by sparse patterns in shallow layers. The non-zero *upper* bound provided by BRL-Attention demonstrates that information *can* flow under bottleneck conditions described in Prop. 2.2.

## F  Datasets and Parameters

Table 9: Statistics of the image/text classification datasets.

| Dataset | Context | Property | Datapoints | Features | Classes |
|---------|---------|----------|------------|----------|---------|
| Mini-ImageNet | Image classification | no graph/$k$-NN graph | 18,000 | 128 | 30 |
| 20News-Groups | Text classification | no graph/$k$-NN graph | 9,607 | 236 | 10 |

Table 10: Statistics of the heterogeneous graph datasets.

| Dataset | Nodes | Node types | Edges | Edge types | Target | Classes |
|---------|-------|------------|-------|------------|--------|---------|
| DBLP | 26,128 | 4 | 239,566 | 6 | author | 4 |
| IMDB | 21,420 | 4 | 86,642 | 6 | movie | 5 |
| ACM | 10,942 | 4 | 547,872 | 8 | paper | 3 |
| Freebase | 43,854 | 4 | 151,034 | 6 | movie | 3 |

Table 11: Training hyperparameters of Standard Full-Transformer in the original LRA paper. Settings are derived from (Tay et al., 2020b).

| Task | Depth | Features | Num Heads | FF size | BSZ | Pooling | LR |
|---|---|---|---|---|---|---|---|
| ListOps | 4 | 512 | 8 | 1024 | 32 | CLS | 5e-2 |
| Text | 6 | 256 | 8 | 1024 | 32 | CLS | 5e-2 |
| Retrieval | 4 | 128 | 4 | 512 | 32 | CLS | 5e-2 |
| Image | 1 | 128 | 8 | 128 | 256 | CLS | 5e-4 |
| Pathfinder | 4 | 32 | 4 | 32 | 512 | Mean | 1e-3 |
| Path-X | 2 | 32 | 4 | 32 | 32 | CLS | 1e-3 |

Table 12: Training hyperparameters of BRL-Former and Transformers tagged with **out-imp** in the LRA experiment. Most settings are similar to that of Vanilla Transformers in (Amos et al., 2023).

| Task | Depth | Features | Num Heads | FF size | BSZ | Pooling | LR |
|---|---|---|---|---|---|---|---|
| ListOps | 6 | 512 | 8 | 1024 | 64 | Mean | 1e-4 |
| Text | 6 | 512 | 8 | 1024 | 64 | Mean | 1e-4 |
| Retrieval | 4 | 128 | 8 | 512 | 16 | Mean | 5e-4 |
| Image | 3 | 64 | 8 | 128 | 16 | Max | 5e-4 |
| Pathfinder | 4 | 128 | 8 | 128 | 16 | Mean | 5e-4 |
| Path-X | 4 | 128 | 8 | 128 | 32 | Max | 5e-4 |

Table 13: Training hyperparameters of BRL-Former in the WikiText-103 modeling. Our settings are similar to that of Vanilla Transformers in `nanoGPT`.

| | Model | Implementation | Depth | Num Heads | Hidden | FF size | w | m | block size | max memory (queue length) | BSZ | LR |
|---|---|---|---|---|---|---|---|---|---|---|---|---|
| Baseline | Full | (Qin et al., 2022a) | 6 | 8 | 512 | 2048 | - | - | 512 | - | 128 | 5e-4 |
| Our-imp | Full | `minGPT/nanoGPT` | 6 | 8 | 512 | 512 * 4 | - | - | 512 | - | 16 * 4 | 5e-4 |
| | Local | | 6 | 8 | 512 | 512 * 4 | 64/128 | - | 512 | - | 16 * 4 | 5e-4 |
| | BRL | | 6 | 8 | 512 | 512 * 4 | 64/128 | 64/128 | 512 | 512 | 16 * 4 | 5e-4 |

Table 14: Training hyperparameters of BRL-Former in the OpenWebText modeling. Our settings are similar to that of Vanilla Transformers in `nanoGPT`.

| Model | Implementation | Depth | Num Heads | Hidden | FF size | w | m | block size | max memory (queue length) | BSZ | LR |
|---|---|---|---|---|---|---|---|---|---|---|---|
| Full | `minGPT/nanoGPT` | 6 | 8 | 512 | 512 * 4 | - | - | 1024 | - | 16 * 5 | 5e-4 |
| Local | | 6 | 8 | 512 | 512 * 4 | 256 | - | 1024 | - | 16 * 5 | 5e-4 |
| BRL | | 6 | 8 | 512 | 512 * 4 | 256 | 256 | 1024 | 1024 | 16 * 5 | 5e-4 |

---

**Algorithm 1** BRL-Former and its Constituent Functions

Given $\mathbf{x}$ (data, $\mathbb{R}^{n \times d}$), $\mathbf{x}_{[\mathtt{ct}]}$ (initialized as $\mathtt{nn.Embeddings}$, $\mathbb{R}^{m \times d_{\mathrm{ct}}}$) and $\mathbf{x}_{\mathrm{hist}}$ (initialized as $\mathbf{0}$, $\mathbb{R}^{n_{\mathrm{hist}} \times d}$).

1: **function** BRL_FORMER($\mathbf{x}, \mathbf{x}_{[\mathtt{ct}]}, \mathbf{x}_{\mathrm{hist}}$, causal)
2:      **if** not causal **then**
3:          $\mathbf{x}_{\mathrm{hist}} \leftarrow \mathtt{concat}(\mathbf{x}_{\mathrm{hist}}, \mathbf{x})[-n_{\mathrm{hist}}:]$
4:      **end if**
5:      **for** $i = 1$ to $L$ **do**
6:          $\mathbf{x}, \mathbf{x}_{[\mathtt{ct}]} \leftarrow \mathtt{brl\_block}(\mathbf{x}, \mathbf{x}_{[\mathtt{ct}]}, \mathbf{x}_{\mathrm{hist}})$
7:      **end for**
8:      **if** causal **then**
9:          $\mathbf{x}_{\mathrm{hist}} \leftarrow \mathtt{concat}(\mathbf{x}_{\mathrm{hist}}, \mathbf{x})[-n_{\mathrm{hist}}:]$
10:      **end if**
11:      **return** $\mathbf{x}, \mathbf{x}_{\mathrm{hist}}$
12: **end function**

1: **function** BRL_BLOCK($\mathbf{x}, \mathbf{x}_{[\mathtt{ct}]}, \mathbf{x}_{\mathrm{hist}}$)
2:      $\mathbf{x}', \mathbf{x}'_{[\mathtt{ct}]} \leftarrow \mathcal{F}_{\mathrm{BRL}}(\mathbf{x}, \mathbf{x}_{[\mathtt{ct}]}, \mathbf{x}_{\mathrm{hist}})$
3:      $\mathbf{h} \leftarrow \mathbf{x} + \mathbf{x}'$
4:      $\mathbf{h}_{[\mathtt{ct}]} \leftarrow \mathbf{x}_{[\mathtt{ct}]} + \mathbf{x}'_{[\mathtt{ct}]}$
5:      $\mathbf{x}_{\mathrm{out}} \leftarrow \mathbf{h} + \mathtt{ffn}(\mathtt{layer\_norm}(\mathbf{h}))$
6:      $\mathbf{x}_{[\mathtt{ct}]}^{\mathrm{out}} \leftarrow \mathbf{h}_{[\mathtt{ct}]} + \mathtt{ffn}(\mathtt{layer\_norm}(\mathbf{h}_{[\mathtt{ct}]}))$
7:      **return** $\mathbf{x}_{\mathrm{out}}, \mathbf{x}_{[\mathtt{ct}]}^{\mathrm{out}}$
8: **end function**

1: **function** $\mathcal{F}_{\mathrm{BRL}}(\mathbf{x}, \mathbf{x}_{[\mathtt{ct}]}, \mathbf{x}_{\mathrm{hist}})$
2:      $\mathbf{x}_{\mathrm{out}} \leftarrow \mathcal{F}_{\mathrm{gen}}(\mathbf{x}; D)$      ▷ Linear generalized attention. We employ Local-Attention with pattern $D_{\mathrm{sw}}$.
3:      $\bar{\mathbf{x}}_{[\mathtt{ct}]} \leftarrow \mathbf{x}_{[\mathtt{ct}]}$
4:      **if** $\mathbf{x}_{\mathrm{hist}}$ is not $\mathbf{0}$ **then**
5:          $\bar{\mathbf{x}}_{[\mathtt{ct}]} \leftarrow \mathcal{F}_{\mathrm{comp}}(\mathbf{x}_{[\mathtt{ct}]}, \mathbf{x}_{\mathrm{hist}}, \gamma)$          ▷ Output in $\mathbb{R}^{n \times d_{\mathrm{ct}}}$
6:      **end if**
7:      $\mathbf{x}_{[\mathtt{ct}]}^{\mathrm{out}} \leftarrow (1 - \beta)\mathtt{layer\_norm}(\bar{\mathbf{x}}_{[\mathtt{ct}]}) + \beta f^M(\mathbf{x}_{[\mathtt{ct}]})$
8:      $\mathbf{x}_{\mathrm{reg}} \leftarrow \sigma_{\mathrm{relu}}(\mathtt{layer\_norm}(\mathcal{F}_{\mathrm{prop}}(\mathbf{x}, \mathbf{x}_{[\mathtt{ct}]}^{\mathrm{out}}, \lambda)))$          ▷ Output in $\mathbb{R}^{n \times d}$
9:      $\mathbf{x}_{\mathrm{out}} \leftarrow \mathbf{x}_{\mathrm{out}} + \mathbf{x}_{\mathrm{reg}}$
10:      **return** $\mathbf{x}_{\mathrm{out}}, \mathbf{x}_{[\mathtt{ct}]}^{\mathrm{out}}$
11: **end function**

1: **function** $\mathcal{F}_{\mathrm{comp}}(\mathbf{x}_{[\mathtt{ct}]}, \mathbf{x}_{\mathrm{hist}}, \gamma)$
2:      $\mathbf{h}^Q \leftarrow \mathtt{split\_heads}(f^Q(\mathbf{x}_{[\mathtt{ct}]}), H)$          ▷ Split into $\mathbb{R}^{H \times m \times (d_{\mathrm{ct}}/H)}$
3:      $\mathbf{h}^{K1}, \mathbf{h}^{K2} \leftarrow \mathtt{split\_heads}(f^{K1,K2}(\mathbf{x}_{\mathrm{hist}}), H)$    ▷ Lin projects to $d_{\mathrm{ct}}$. Each split into $\mathbb{R}^{H \times n_{\mathrm{hist}} \times (d_{\mathrm{ct}}/H)}$
4:      $\mathbf{h}^V \leftarrow \mathtt{split\_heads}(f^V(\mathbf{x}_{\mathrm{hist}}), H)$          ▷ Lin projects to $d_{\mathrm{ct}}$. Split into $\mathbb{R}^{H \times n_{\mathrm{hist}} \times (d_{\mathrm{ct}}/H)}$
5:      $\mathbf{S}^{K1}, \mathbf{S}^{K2} \leftarrow \sigma_{\mathrm{attn}}(\mathbf{h}^Q(\mathbf{h}^{K1})^\top), \sigma_{\mathrm{attn}}(\mathbf{h}^Q(\mathbf{h}^{K2})^\top)$          ▷ In $\mathbb{R}^{H \times m \times n_{\mathrm{hist}}}$
6:      $\mathbf{x}_{[\mathtt{ct}]}^{\mathrm{out},h} \leftarrow (\mathbf{S}^{K1,h} - \gamma \mathbf{S}^{K2,h})\mathbf{h}^{V,h}$          ▷ Each head-wise embedding has shape in $\mathbb{R}^{m \times (d_{\mathrm{ct}}/H)}$
7:      $\mathbf{x}_{[\mathtt{ct}]}^{\mathrm{out}} \leftarrow \mathtt{reshape}(\mathtt{group\_norm}(\{\mathbf{x}_{[\mathtt{ct}]}^{\mathrm{out},h}\}_{h=1}^H))$          ▷ Group norm and reshaped into $\mathbb{R}^{m \times d_{\mathrm{ct}}}$
8:      **return** $\mathbf{x}_{[\mathtt{ct}]}^{\mathrm{out}}$
9: **end function**

1: **function** $\mathcal{F}_{\mathrm{prop}}(\mathbf{x}, \mathbf{x}_{[\mathtt{ct}]}, \lambda)$
2:      $\mathbf{z}^Q \leftarrow \mathtt{split\_heads}(f^\phi(\mathbf{x}), H)$          ▷ In shape $\mathbb{R}^{H \times n \times (d/H)}$
3:      $\mathbf{z}^K, \mathbf{z}^V \leftarrow \mathtt{split\_heads}(f^{K,V}(\mathbf{x}_{[\mathtt{ct}]}), H)$          ▷ Lin projects to $d$. Output in shape $\mathbb{R}^{H \times m \times (d/H)}$
4:      $\mathbf{S} \leftarrow \sigma_{\mathrm{attn}}(\mathbf{z}^Q(\mathbf{z}^K)^\top)$          ▷ In $\mathbb{R}^{H \times n \times m}$
5:      $\mathbf{x}_{\mathrm{out}}^h \leftarrow \lambda \mathbf{S}^h \mathbf{z}^{V,h}$          ▷ Each head-wise embedding has shape in $\mathbb{R}^{n \times (d/H)}$.
6:      $\mathbf{x}_{\mathrm{out}} \leftarrow \mathtt{reshape}(\{\mathbf{x}_{\mathrm{out}}^h\}_{h=1}^H)$          ▷ Reshaped into $\mathbb{R}^{n \times d}$
7:      **return** $\mathbf{x}_{\mathrm{out}}$
8: **end function**

---

