# OpenReview forum: "Toward Linearly Regularizing the Geometric Bottleneck of Linear Generalized Attention"
_TMLR — Accepted by TMLR_

### Review · Reviewer_Av48 · 2025-04-28

**Summary Of Contributions:**

The paper presents BRL-Attention (Bottleneck Regularized Linear Attention), a novel attention scheme to counteract inefficiencies in existing efficient transformer models. Traditional full-attention Transformers enjoy quadratic complexity, while recent efficient approaches — pattern-based (e.g., sparse or local attention) and kernel-based (e.g., low-rank approximations) — fall short either in having bottlenecks or sacrificing accuracy. BRL-Attention enhances pattern-based attention with a limited set of compressed tokens as a global information pool, enabling long-range interaction without quadratic cost and without sacrificing causal masking (key to autoregressive usage). Theoretical inspection and extensive experimentation on long-sequence benchmarks (e.g., Long Range Arena, WikiText-103, large graph corpora) attest that BRL-Attention is capable of competing or beating full-attention transformers with significant reductions in memory and compute cost.

**Audience:**

Yes

**Broader Impact Concerns:**

There is no concern.

**Claims And Evidence:**

Yes

**Requested Changes:**

* Describe in greater detail how sensitive BRL's performance is to compressed tokens' numbers, $\\\\beta,\\\\gamma$, etc.

* Include experiments beyond text and graphs, e.g., image or speech processing (even some early ones).

* Enumerate explicitly any modifications of re-implemented baselines (e.g., "Transformer (our-imp)") so that readers can have confidence that improvements are genuine.

* Include visualization showing how token representations evolve with and without BRL in deeper layers.

* Some figures (e.g., Fig. 4, Fig. 5) are too busy and may be more readable with a cleaner layout.

* Include sketch proof diagrams in the main paper (Appendix is okay for full proofs, but the main paper needs to give a summary of key steps naturally).

**Strengths And Weaknesses:**

# Strengths

* The idea of combining sparse attention with compressed global tokens in theoretically sound ways is novel.

* Bottleneck regularization is well-justified and extensively explored.

* Formal sensitivity analysis and bottleneck evaluation based on Jacobians.

* Proofs of expressivity equivalence to full attention under the BRL setup.

* Tests on both sequence modeling and graph node classification tasks.

* Exhaustive ablations (effect of compressed tokens, attention distinction).

* Recorded substantial improvements in memory and speed, particularly for extremely long sequences (up to $2^18$).

* In contrast to some kernel-based methods, BRL maintains causal masking, and therefore can be used both for encoder and decoder scenarios.

* The organization of sections makes the method simple to trace despite technical complexity.

# Weaknesses

* Heavily Dependent on Engineering Decisions: There are several hyperparameters (e.g., window size, number of compressed tokens, $\lambda,\gamma$) that need to be tuned, and the method's robustness to these choices is not examined thoroughly. How general is the design to other domains (e.g., vision tasks)?

* Possible Complexity in Implementation: Newly introduced elements (compression and propagation mappings) complicate training pipelines compared to baseline attention surrogates like Performer.

* Comparisons could be extended: The paper mentions "our-implementation" of some baselines (e.g., full attention) — more details on deviations from vanilla baselines would avoid fairness concerns. No experiments on machine translation or image classification tasks, where causal models are important and the gains would be vital.

* Insufficient qualitative analysis: No qualitative visualizations of how BRL-Attention alleviates bottlenecks compared to local attention over multiple layers (e.g., information flow diagrams beyond Figure 1).

---

> ### Author Response · Authors · 2025-05-13
> **Response to Reviewer Av48: Part 1**
>
> We thank the reviewer for your thorough review and insightful comments on our manuscript. All the changes made to the manuscript are highlighted in blue. Below, we address each of the concerns in detail.
>
> ---
>
> ## Questions
>
> ```
> W1, RC1: Heavily Dependent on Engineering Decisions: There are several hyperparameters...Describe in greater detail how sensitive BRL's performance...
> ```
> **Response:** We thank the reviewer for this suggestion. We want to clarify that our method is not very hyperparameter dependent. In the revised manuscript, we have included a more detailed discussion on initialization of parameters in Section 2.5. Specifically:
>
> - Empirically, we find setting $\beta=0.5$ (which is not learnable) performs well generally, while set $\beta=0$ offers negative impact (Fig. 7a).
> - For sensitivity to compressed tokens $m$, we conducted additional experiment on OpenWebText (Appendix A.3) where we tested 128~256 compressed tokens under 1024 block size decoder-only model. Similar to the results in Fig.7, a larger number of compressed token would benefit the performance, while $\le 256$ is enough for achieving outstanding performance against Full-Attention.
> - On **learnable** parameter $\lambda$, we find the initialization does not significantly impact the final performance (0.1, 0.5, 1.0) as demonstrated in Fig.7a. However, a smaller $\lambda$ would benefict the optimization, *i.e.* $\lambda \in [0.1, 0.5]$.
> - On **learnable** parameter $\gamma$, we set the initial value of it to $0$ to facilitate a cold start, such that theoretically it would not perform worse than a $\mathcal{F}_\mathrm{comp}$ without M1, as detailed in Section 2.4. Meanwhile, we find that disabling the learnbaility of $\gamma$ offers negative impact (Fig.7b), this verifies the importance of $\gamma$ to exist, as well as setting it to be learnable.
>
> ---
>
> ```
> W2, RC5: Possible Complexity in Implementation... Some figures are too busy
> ```
> **Response:** We thank the reviewer for the critics on implementation complexity and readability. In the revised manuscript:
> - We have restructured the method section to explicitly label subsections with "[Practice]" and "[Theory]" to delineate the practical implementation aspects from the theoretical underpinnings, making it easier for readers to follow the design choices.
> - We introduce Research Questions at the beginning of Sections 2.3 and 2.4 to guide the reader and improve the flow from motivation to practical design and theoretical justification.
> - We compressed some content and moved some content to appendix: e.g. We condensed Proposition 2.6, we moved the algorithm to appendix.
> - We significantly simplified the Figure 4 to make it more readable.
> - We provided a Sketch Proof for Theorem 2.9 following the request of reviewer Av48.
>
> ---
>
> ```
> W3: more details on deviations from vanilla baselines would avoid fairness concerns
> ```
> **Response:** We have revised Section 3.2, Table 13, Table 14 and added Table 11 to appendix to enhance the detail on implementation difference of compared baselines.
>
> ---
>
> ```
> W3, R2: Include experiments beyond text and graphs, e.g., image or speech processing (even some early ones)...
> ```
> **Response:** We appreciate the reviewer's suggestion to explore the applicability of BRL-Attention to other domains.
>
> - **Regarding image processing:** As we have elaborated on this in our limitations (Appendix D.5). Our analysis suggests that vision tasks employing models like Vision Transformers (ViTs), where the total number of tokens is often small may not substantially benefit from BRL-attention's advantages over standard full attention. For instance, the total tokens of vit for a single pass in ViT-B/16 is 197 tokens. Already, a typical sliding window $w=128$ is approximately the length of whole sequence. According to the additional memory/time complexity we provide in Table 6, Appendix A.2: or sequence lengths $n\le 512$, the memory and time consumption of local attention do not markedly differ from those of full attention. As the receptive field and complexity both does not change significantly comparing local to full-attention, we can state that there is no attention bottleneck on local-attention on vision tasks. Therefore, we do not anticipate that BRL-Attention would offer a notable improvement over full attention.
> - **Concerning speech processing:** We recognize the importance of this domain. However, conducting experiments in speech processing was not feasible for this revision due to two primary factors: our current team's expertise does not deeply cover this specialized area, and the pre-training of effective speech models typically demands substantial computational resources that were beyond our immediate access. We have acknowledged this as a limitation and a promising direction for future research in Appendix D.5.

---

> ### Author Response · Authors · 2025-05-13
> **Response to Reviewer Av48: Part 2**
>
> ```
> W4, RC4: No qualitative visualizations of how BRL-Attention alleviates bottlenecks compared to local attention over multiple layers ...
> ```
> **Response:** We agree that such a visualization would be beneficial. We have included a new visualization in the revised manuscript Appendix C, Figure 14, page 20 about how BRL-Attention enhances information flow compared to local attention. For context, the visualization depicts:
>
> 1. **Full-Attention:** The standard full attention score matrix.
> 2. **Local-Attention:** The attention scores confined to the sliding window.
> 3. **BRL-Attention:** We showcase three key components:
>     - The scores from its local attention part.
>     - The Reconstructed Regularizer Score (RRS), which represents the global, long-range dependencies captured via our compressed tokens.
>     - The final reconstructed BRL-Attention score (not the actuall BRL-attention score since direct measuremen is difficult), which is the weighted sum (by coefficient $\lambda$) of the local scores and the RRS. We detail the entry i~j of RSS in Eq. 15.
>
> We observe that the Reconstructed Regularizer Score (RRS) encapsule the long range dependencies of tokens for each query. We observe that the Reconstructed Regularizer Score (RRS) encapsule the long range dependencies of tokens for each query. In Figure 14, we can observe that for some queries, the keys to attend (vertical lines) align with that of the full-attention. This indicate that the regularizer is learning useful information from global context. By incorporating this global context (which is appended to layer-wise embedding), the BRL-Attention overcomes the receptive field limitations inherent in purely local attention.
>
> ---
>
> ```
> RC6: Include sketch proof diagrams in the main paper...
> ```
> **Response:** We appreciate and agree with the reviewer's suggestion to include sketch proofs in the main paper to enhance the accessibility. In line with this, we have incorporated a sketch proof for Theorem 2.9 in the main paper. We selected Theorem 2.9 for this treatment as it involves a more extensive, multi-step deductive argument where a high-level summary of key steps is particularly beneficial for the reader.
>
> For our other propositions and theoretical results, their proofs are generally more direct, often consisting of derivations or computational steps rather than intricate logical chains. Also considering the page limit for the main paper, we focused on providing the sketch proof for the multi-step Theorem 2.9 to best utilize the available space.
>
> ---
>
> We have substantially improved the manuscript and, hopefully, we addressed the concerns raised. We thank the reviewer again for their valuable time and constructive feedback.

---

### Review · Reviewer_jtaY · 2025-05-03

**Summary Of Contributions:**

This paper considers a new attention mechanism called BRL-attention, with the goal of combining the strength of both pattern-based sparse attention, and linear attention. The major issue of pattern-based attention, especially the variant of sliding window, restrict long range dependence between two points after a certain number of attention layers. Linear attention on the other hand has a significant gap between theoretical and practical runtime due to limited parallelization. The proposed method is a variation that allows pattern-based sparsity for attention, while ensure the long range dependency to be captured via a regularizer term. The regularization crucially uses feature mapping, as it selects a subset of compressed tokens, compute the feature mapping as the queries, and project the compressed tokens as keys and values, then compute the attention with these components.

The main theoretical result is that, if we let attention bottleneck be that the Jacobian at the $l$-th layer of the $i$-th token with respect to the $p$-th token at the initial layer is roughly 0, then one could prove that the sliding window is bottlenecked unless the number of attention layers is large enough. Under the assumption that the feature mapping used in regularization approximates the random feature mapping and some requirement on the embedding dimension of the compressed tokens, then the attention between feature mapped queries and compressed tokens key/values actually approximates the self-attention module. Assuming the tokens and weight matrices are bounded in norms, they subsequently show that an MLP with $O(1)$ layers could approximate random feature mapping, providing a practical solution to design the feature mapping. Finally, they show that under proper scaling, the cross attention term for the regularization has small norm, and the BRL attention can approximate the full attention.

Empirically, authors show that their algorithm is memory and time efficient, and obtains good metrics in various tasks.

**Audience:**

Yes

**Claims And Evidence:**

Yes

**Requested Changes:**

Major:

1. This paper has a lot of notations that are used before defined, or are never defined. For example, in Eq. (5), LN is used without ever defined, I think that's LayerNorm? Also, what is $\sigma$ and $\odot$? I assume $\sigma$ is ReLU, and $\odot$ is elementwise product? While these notations are introduced at the beginning of Section 2, but they are only explained in Section 2.3, and $\odot$ is never explained. Authors should at least include a preliminary section in the appendix, or explain these notations directly after they are used.

2. I think authors should not mix math notations with pytorch. For example, in Eq. (7), authors should either adopt math notations for both (which I think is a better option), or use pytorch syntax. Also, it is quite unusual to write algorithm block in pytorch instead of pseudocode. If space is a concern, authors could defer the algorithm block in the appendix, but I do feel it might be better to write in pseudocode.

3. It's important to explicitly spell out the connection between regularization term and alleviation of bottleneck. In particular, in addition to a sensitivity upper bound on the regularization term, it's important to also exhibit a nontrivial lower bound. In fact, I'm a bit confused in the statement of Proposition 2.8 where the upper bound means BRL is "repellent" to the bottleneck, I think you need a lower bound to reach the conclusion. This is particularly crucial since most of the theories in this paper are not deep and serve as explanatory purpose, and a lower bound on the regularization would ensure that, at least in the case of sliding window, the bottleneck is actually alleviated. Ideally, the lower bound should not scale with $1/n$, as one would expect the context length to be long (in fact, this is the main motivation to design efficient attention approximation or new architecture).

4. I'm not an expert on experiments, I'll defer comments on the experiments to other reviewers with more expertise on this front.


Minor:

- paragraph above the captions for Table 1: "performs well differently from the training time performance", not sure what the authors try to express here, please clarify.

**Strengths And Weaknesses:**

Strength: This paper proposes a new architecture that combines sparse attention and linear attention. The regularization term novelly combines the feature mapping used in linear attention and a set of carefully-crafted compressed tokens, to compute a small cross-attention term. This helps to alleviate the issue of long range dependence issue called by sparse attention, particularly the variant of sliding window. To theoretically capture this notion, the paper develops a notion of bottleneck which is defined as the Jacobian between two points across layers is approximately 0. They prove that sliding window indeed causes bottleneck unless the transformer has large enough attention layers. The regularization is introduced to address this issue, and they prove that indeed the regularization term does not introduce too many errors, and the BRL attention approximates the vanilla full attention. Experiments are performed to verify the new architecture works well in practice.

Weakness: There are two main weaknesses.

1. The connection between regularization and alleviation is not strong enough. The main selling point of the paper is that, since pattern-based attention is provably bad at capturing long range dependence in terms of Jacobian bottleneck. Hence, the regularization term is very carefully designed to address this issue. However, the only part mentioning the *theoretical* effect of regularization is a very short paragraph saying that the regularization term makes the sensitivity bound nonzero, and this implies that the information is more likely to flow from token $i$ to $p$ across layers. This connection is also not made clear in the appendix. It is worth noting that most theories in this paper are explanatory and not very deep, they mainly serve the purpose of providing explanations on the new architecture, so it's crucial for authors to articulate clearly how the regularization term helps to alleviate the bottleneck.

2. The structure of the paper is a bit messy. Authors attempt to compress many contents including theoretical and empirical into 12 pages, so I do understand the difficulty, but this results in the paper requires multiple re-readings in order for me to fully understand. For example, I feel Section 2.4, **Instantiation of Information Compression** could be further condensed, as the main goal is to show that the regularization term is not too large? More for this part in requested changes.

---

> ### Author Response · Authors · 2025-05-13
> **Response to Reviewer jtaY: Part 1**
>
> We thank the reviewer for the thorough review and constructive comments. We have addressed the identified weaknesses and requested changes in the revised manuscript as detailed below.
>
> ---
>
> ## Questions
>
> ```
> W1, RC3: The connection between regularization and alleviation is not strong enough... It's important to explicitly spell out the connection between regularization term and alleviation of bottleneck. In particular, in addition to a sensitivity upper bound on the regularization term, it's important to also exhibit a nontrivial lower bound...
> ```
> **Response:** You've rightly pointed out the importance of demonstrating how the BRL-Attention mechanism overcomes the limitations of pattern-based sparse attention, specifically the Jacobian bottleneck issue. Our core argument is that the regularization term ensures a *possibility* of information flow where it was previously structurally guaranteed to be zero under certain conditions for sparse attention alone.
>
> 1.  **Alleviating the "Zero-Sensitivity" Bottleneck:**
>     Proposition 2.2 in our paper demonstrates that for a sliding window attention pattern ($D_{sw}$), the sensitivity of the $i$-th token at layer $l$ with respect to the $p$-th token at the initial layer (i.e., the Jacobian $\frac{\partial(\mathcal{F}_{gen}^{(l)}(x^{(l)};D))_{i}}{\partial x_{p}^{(0)}}$) is **strictly zero** if the number of layers $l$ is less than a threshold determined by the distance $M$ between tokens and the window size $w$ (specifically, $l < \lceil\frac{2M}{w-1}\rceil-1$). This signifies a hard bottleneck: no information can propagate between these tokens through the generalized attention mechanism $\mathcal{F}_{gen}$ alone within that depth.
>
>     Our BRL-Attention introduces a regularization term, $\mathcal{F}_{prop}$. As stated in the discussion following Proposition 2.7 in the revised manuscript, the sensitivity bound associated with BRL-Attention, $|\frac{\partial\mathcal{F}_{prop}^{(l)}(x^{(l)};x_{[ct]}^{(l+1)})_{i}}{\partial x_{p}^{(l)}}|$, has a non-zero upper bound given by $\mathcal{O}(\frac{r_{x}r_{ct}r_{W}^{6}}{\sqrt{dd_{ct}}}(1+|\gamma|))$. This bound is applicable for $l \in [0, L]$. By adding the regularization, the overall BRL-Attention mechanism, $\mathcal{F}_{BRL} = \mathcal{F}_{gen} + \mathcal{F}_{prop}$, now has a path for information propagation whose sensitivity is **not necessarily zero**, even for small $l$ where $\mathcal{F}_{gen}$ alone would be bottlenecked. This transition from a *guaranteed zero sensitivity* to a *potentially non-zero sensitivity* is what we mean by "alleviation" of the bottleneck. The regularizer provides an alternative pathway for information, ensuring that distant tokens *can* interact even within a few layers.
>
> 2.  **Regarding a "Non-trivial Lower Bound":**
>     We appreciate the suggestion to exhibit a non-trivial lower bound on the regularization term's effect to further strengthen the argument. However, we believe that establishing a universally non-trivial (i.e., strictly greater than zero) lower bound for sensitivity (information propagation) is generally problematic, even for standard full-attention mechanisms.
>
>     The actual flow of information, and thus the Jacobian, is inherently data-dependent and context-specific:
>     - **Data Specificity:** If, for a particular input sequence, there is no meaningful semantic relationship to be captured between two specific tokens $i$ and $p$, the attention scores connecting them (even in a full-attention model) might naturally be zero or near-zero. In such an instance, the Jacobian representing the influence of token $p$ on token $i$ would also be zero, leading to a trivial (zero) lower bound.
>     - **On Full-Attention w/o Bottleneck:** Even in a Full-Attention Transformer, if the attention mechanism learns that token $i$ should not attend to token $p$ for a given context, the corresponding attention weight will be zero, and thus the propagated information (and its gradient) through that specific path will be zero.
>
>     Therefore, a lower bound on sensitivity would likely still be zero in many valid scenarios. The crucial aspect of our BRL-Attention is not to guarantee a *minimum amount* of information flow between all token pairs at all times, but rather to remove the *structural certainty of zero flow* imposed by sparse patterns in shallow layers. The non-zero *upper* bound provided by BRL-Attention demonstrates that information *can* flow under bottleneck conditions described in Proposition 2.2.
>
> We hope this clarifies how our regularization addresses the bottleneck issue and explains our perspective on lower bounds for information propagation in attention mechanisms. We are happy to incorporate further clarifications into the paper if necessary.

---

> > ### Author Response · Authors · 2025-05-13
> > ***Fixing the Latex rendering error of response: "Alleviating the "Zero-Sensitivity" Bottleneck"**
> >
> > **Alleviating the "Zero-Sensitivity" Bottleneck:**
> >
> > Proposition 2.2 in our paper demonstrates that for a sliding window attention pattern ($D_{sw}$), the sensitivity of the $i$-th token at layer $l$ with respect to the $p$-th token at the initial layer (i.e., the Jacobian $\partial F_{gen}(x; D )i/ \partial x_p)$ is **strictly zero** if the number of layers $l$ is less than a threshold determined by the distance $M$ between tokens and the window size $w$ (specifically, $l < \lceil\frac{2M}{w-1}\rceil-1$). This signifies a hard bottleneck: no information can propagate between these tokens through the generalized attention mechanism $\mathcal{F}_{gen}$ alone within that depth.
> >
> > Our BRL-Attention introduces a regularization term, $F_{prop}$, as stated in the discussion following Proposition 2.7 in the revised manuscript, the sensitivity bound associated with BRL-Attention: $\partial F_{prop}^{(l)}(x^{(l)};x_{[ct]}^{(l+1)})i / \partial x_{p}^{(l)}$ has a non-zero upper bound given by $\mathcal{O}(\frac{r_{x}r_{ct}r_{W}^{6}}{\sqrt{dd_{ct}}}(1+|\gamma|))$. This bound is applicable for $l \in [0, L]$. By adding the regularization, the overall BRL-Attention mechanism, $F_{BRL} = F_{gen} + F_{prop}$, now has a path for information propagation whose sensitivity is **not necessarily zero**, even for small $l$ where $\mathcal{F}_{gen}$ alone would be bottlenecked. This transition from a *guaranteed zero sensitivity* to a *potentially non-zero sensitivity* is what we mean by "alleviation" of the bottleneck. The regularizer provides an alternative pathway for information, ensuring that distant tokens *can* interact even within a few layers.

---

> > > ### Comment · Reviewer_jtaY · 2025-05-27
> > >
> > > I thank authors for the response, overall I'm quite satisfied with the modifications. For the lower bound on the regularization term, I understand authors' reasoning, but I would strongly recommend to prove a lower bound even under a set of restrictive assumptions, or incorporate the response on why alleviating the zero-sensitivity bottleneck into the paper. Page limit is certain an issue -- maybe it makes sense to add a section in appendix to address this?

---

> > > > ### Author Response · Authors · 2025-05-28
> > > > **Added new section in appendix about lower-bound**
> > > >
> > > > We thank the reviewer for the suggestion. We aggree that incorporating a justification on lower-bound is necessary for the audience.
> > > >
> > > > In the revised manuscript, we have added a section in appendix E.7 (page 31~32) about the lower-bound of the sensitivity analysis. In specific:
> > > > - We show a proof with simplified architecture that: assuming the smallest singular values of weight matrices $s_W$ to be greater than zero, then the jacobian of BRL sensitivity $\ge \frac{m s_W^6}{d} > 0$.
> > > > - We added a discussion below the proof noting that establishing a universally *strictly greater than zero* lower bound for sensitivity is generally problematic without any assumption.
> > > > - We added a reference in Proposition 2.7 to Appendix E.7.
> > > >
> > > > We hope these revisions address the reviewer's concerns.

---

> > > > > ### Comment · Reviewer_jtaY · 2025-05-28
> > > > >
> > > > > I thank the authors for the response and revisions, my concerns are addressed.

---

> ### Author Response · Authors · 2025-05-13
> **Response to Reviewer jtaY: Part 2**
>
> ```
> W2: The structure of the paper is a bit messy...
> ```
> **Response:** We acknowledge the density of the paper due to the page limit. As also mentioned in our response to Reviewer Pfto, Section 2 has been significantly restructured and we have condensed some unimportant information in both Sections 2.3 and 2.4. In summary:
>
> - We have restructured the method section to explicitly label subsections with "[Practice]" and "[Theory]" to delineate the practical implementation aspects from the theoretical underpinnings, making it easier for readers to follow the design choices.
> - We introduce Research Questions at the beginning of Sections 2.3 and 2.4 to guide the reader and improve the flow from motivation to practical design and theoretical justification.
> - We compressed some content and moved some content to appendix: *e.g.* We condensed Proposition 2.6, moved the algorithm to appendix and simplified the Figure 4 to make it more readable.
> - We provided a Sketch Proof for Theorem 2.9 following the request of Reviewer Av48.
>
>
> ---
>
> ```
> RC1: This paper has a lot of notations that are used before defined, or are never defined...
> ```
> **Response:** We apologize for any lack of clarity in notation. In the revised manuscript (Section 2), we have ensured notations are defined upon their first use.
>
>
> ---
>
> ```
> RC2: I think authors should not mix math notations with pytorch... f space is a concern, authors could defer the algorithm block in the appendix, but I do feel it might be better to write in pseudocode...
> ```
> **Response:** We thank the reviewer for this advice. In the revised manuscript, we have modified Eq.7 to adopt the unified math notation, as well as the surrounding text to make it more consinstent. For the algorithm block (originally next to Figure 4), we have deferred it to the Appendix (last page) and revised it to pure pseudocode format instead of pytorch format for better clarity and universality.
>
>
> ---
>
> ```
> Minor1: paragraph above the captions for Table 1...not sure what the authors try to express here...
> ```
> **Response:** We apologize for the unclear phrasing. This sentence referred to the Performer model. In the revised manuscript, this has been clarified to: "The Performer, different from its autoregressive training performance (worse than BRL-Attention on both time and memory cost when scaling to longer sequences), performs well on inference-speed." The intended meaning is that while Performer might have challenges in parallelizing during autoregressive training (leading to slower training than its linear inference complexity might suggest), its inference speed is indeed efficient and aligns better with its theoretical linear complexity, which is what we observed.
>
> ---
>
> We believe these revisions and explanations address the reviewer's concerns and significantly clarify the contributions and theoretical underpinnings of our work. We hope Reviewer jtaY finds these changes satisfactory.

---

### Review · Reviewer_Pfto · 2025-05-05

**Summary Of Contributions:**

This paper introduces BRL-Attention, a novel attention mechanism for Transformers that achieves linear complexity while maintaining strong performance. The authors combine local and global attention to tokens that compress context, justifying this design with a theoretical analysis of attention layer expressivity.

**Audience:**

Yes

**Claims And Evidence:**

Yes

**Requested Changes:**

I would like to see all the weaknesses mentioned above addressed before recommending acceptance. In particular, this means:
- improving the presentation of the paper, by making the motivation of the architecture and its description more accessible.
- providing experimental evidence that BRL attention is much cheaper than standard attention on the different tasks and that there are reason to believe that it will scale decently well.
- integrating a thorough discussion of the deep SSMs literature where adequate.

**Strengths And Weaknesses:**

**Strengths**

1. The paper addresses an important challenge in the field: finding alternatives to traditional Transformers that maintain performance while reducing their quadratic complexity problem.
2. The theoretical foundation is robust, with analysis of local attention limitations and kernel approximation techniques that directly inform the model's architecture.
3. The empirical validation is comprehensive, comparing BRL-attention across multiple modalities against several efficient Transformer alternatives. The results demonstrate consistent improvements over standard attention mechanisms, particularly in handling long-range dependencies.


**Weaknesses**
1. The architecture is rather complex, which has several consequences:
  - The intricate design may deter other researchers from implementing or building upon this work.
  - Key illustrations (Algorithm 1 and Figure 4) lack clarity and accessibility.
  - Having this many components may increase the computational overhead despite linear scaling (more detail on that in weakness 3).
2. The theoretical section would benefit from clearer connections to the practical implementation. Currently, the technical results appear overly technical from the architecture choices, and fail to make the motivation of the design clear to the readers.
3. While the authors study the memory and the speed of BRL-attention on a toy task, it is not clear whether the same trends still hold in the experiments of Section 3.2 to 3.4. Given the modest performance gains over standard attention, it is important that the costs of BRL are much lower than the ones of attention for it to be of practical interest.
4. The relatively small scale of experiments raises questions about how BRL-attention would perform in large-scale deployments, where the advantages of Transformers on more efficient alternatives tend to become more pronounced.
5. Comparison to the deep state-space model literature is missing. It tackles the same problem as this paper, has been thoroughly investigated in the last few years, and achieves competitive results. For example, S4 significantly beat all the methods reported on the LRA benchmark.

---

> ### Author Response · Authors · 2025-05-13
> **Response to Reviewer Pfto**
>
> We thank the reviewer for the insightful feedback. We have considered all comments and have revised the manuscript accordingly. All the changes made to the manuscript are highlighted in blue. Below, we address each of the concerns in detail.
>
> ---
>
> ## Questions
>
>
> ```
> W1.1, W2: The intricate design may deter other researchers from implementing... The theoretical section would benefit from clearer connections to the practical implementation...
> ```
>
> **Response:** We acknowledge the reviewer's concern regarding the perceived complexity. In the revised manuscript, we have restructured the method section to explicitly label subsections with "[Practice]" and "[Theory]" to delineate the practical implementation aspects from the theoretical underpinnings, making it easier for readers to follow the design choices.
>
> Also, we have added RQ guildlines to Sections 2.3 and 2.4 to facilitate a clearer motivation for the readers:
> - [Section 2.3]: "How can a small, trainable set of global memory capture long-range context in linear complexity while preserving the expressibility."
> - [Section 2.4]: "Given a squashed pattern D, how a concrete compression mapping lets the F_gen approximate full attention yet keep the linear complexity and offers extended sensitivity bound."
>
> ---
>
> ```
> W1.2: Key illustrations (Algorithm 1 and Figure 4) lack clarity and accessibility.
> ```
> **Response:** In the revised manuscript, we have simplified Figure 4 for better presentation. Nonetheless, we rewrote the Algorithm 1 from pytorch-like to pure pseudo code for better clarity and consistency, then moved Algorithm 1 to the appendix.
>
> ---
>
> ```
> W1.3, W3: Having this many components may increase the computational overhead despite linear scaling ... it is not clear whether the same trends still hold in the experiments of Section 3.2 to 3.4
> ```
> **Response:** We agree that demonstrating cost-effectiveness on real tasks is crucial. In the revised manuscript:
>
> - We have added a new Appendix A.2 titled "Complexity Evaluation on Real-World Textual Dataset". This section provides detailed memory (Table 6) and time efficiency (Table 7) comparisons on the WikiText-103 dataset (used in Section 3.2) for various sequence lengths (n=128 to n=4096). These tables clearly show that BRL-Attention is significantly cheaper than standard attention on this real-world language modeling task. For instance, in Table 6 (autoregressive training), at n=2048, Full-Attention uses 12.772GB, while BRL-Attention (w=128,m=128) uses 6.428GB. At n=4096, Full-Attention is OOM, while BRL-Attention uses 9.615GB. The inference times in Table 7 also show substantial speedups.
> - We would also like to note that, in the graph modeling tasks, Figure 8b also includes a memory comparison, showing BRL-Attention's efficiency ("Memory Cmp."). While performance gains over full attention are sometimes modest on graph tasks, they come with significant computational savings.
>
>
> ---
>
> ```
> W4, RC2: The relatively small scale of experiments raises questions about how BRL-attention would perform in large-scale deployments...
> ```
> **Response:** We acknowledge the desire for larger-scale experiments. In the revision, we have conducted new experiment in Appendix A.3, where we added experiments on the OpenWebText dataset, which is substantially larger than WikiText-103 (~9B tokens, taking up ~17GB of disk after preprocessing). Figure 10 shows training and validation loss curves for Full-Attention, Local-Attention, and BRL-Attention (m=w=128 and m=w=256) on OpenWebText with a block size of 1024.
>
> The results in Figure 10 demonstrate that BRL-Attention continues to perform competitively or better than Full-Attention, and significantly better than Local-Attention on OpenWebText. This suggests that the benefits of BRL-Attention in mitigating attention bottlenecks and improving performance hold as the dataset scale increases. The linear complexity scaling demonstrated in Appendix A.2 further supports its suitability for larger deployments.
>
>
> ---
>
> ```
> W5, RC3: Comparison to the deep state-space model literature is missing... integrating a thorough discussion of the deep SSMs literature where adequate.
> ```
> **Response:** We thank the reviewer for pointing out this important line of related work. We have now included a dedicated discussion on SSMs in Appendix D.4.2, where we gave brief review on the SSM family, architectural contrasts between BRL-Attention and SSMs, and discussed the fairness of direct comparison upon purposes of two models.
>
>
> ---
>
>
> We believe these revisions address the reviewer's concerns. We hope the reviewer finds these changes satisfactory.

---

> > ### Comment · Reviewer_Pfto · 2025-05-26
> >
> > I appreciate the authors' answer and acknowledge that their changes have solidified the paper. I am overall satisfied with the answer. However, there is one last point that the authors fail to address, which the hybrid model literature (part of the layers are SSMs / recurrent layers, part of them are Transformer layers). Given that the proposed approach mixes the two paradigms within a single layer (whereas these hybrid architectures mix them at the layer level), a thorough discussion of the benefits and weaknesses of each approach would be valuable. Here are a few references that may be useful:
> > - Griffin https://arxiv.org/pdf/2402.19427
> > - https://www.nature.com/articles/s41598-024-55483-x
> > - https://arxiv.org/abs/2502.10807
> > - https://arxiv.org/abs/2403.19887
> > - https://arxiv.org/html/2403.18063v2

---

> > > ### Author Response · Authors · 2025-05-27
> > > **Added discussion on hybrid model to Appendix D.4.3**
> > >
> > > We thank the reviewer for the suggestion.
> > >
> > > We have added a section discussing the related work in Appendix D.4.3 (page 25~26) in the revised manuscript. We divide the literature of Transformer hybrid architecture as (1) *Transformer-Recurrent Hybrids* e.g. Griffin, FLASH; (2) *Transformer-SSM Hybrids* e.g. Jamba, HybriDNA; and (3) *Attention-Centric Hybrids* e.g. our approach. For each catagory we discussed the pros and cons.
> > >
> > > We hope that our updated version of the manuscript address your concerns.

---

> > > > ### Comment · Reviewer_Pfto · 2025-05-27
> > > >
> > > > Thank you for your answer. All my concerns that could have been reasonably addressed have been addressed.

---

### Decision · Action_Editor_qNTH · 2025-06-20

**Recommendation:** Accept as is

**Audience:**

Yes

**Audience Explanation:**

Efficient attention is a critical area for modern machine learning and architecture research, findings in this paper are relevant to both researchers and practitioners in the area.

**Claims And Evidence:**

Yes

**Claims Explanation:**

This paper proposes a novel attention architecture that addresses the limitations of both sparse attention and linear attention. All reviewers have agreed that the empirical results are extensive and solid, and the theoretical results are interesting.